# Neural Tangent Kernel Perspective on Parameter-Space Symmetries

**Ori Shem-Ur , Khen Cohen , Yaron Oz**
School of Physics and Astronomy
Tel Aviv University
Tel Aviv-Yafo, 6997801, Israel
`{orishemor,khencohen}@mail.tau.ac.il, yaronoz@tauex.tau.ac.il`

## Abstract

Deep learning models, such as wide neural networks, can be viewed as nonlinear dynamical systems composed of numerous interacting degrees of freedom. When such systems approach the limit of infinite number of degrees of freedom, their dynamics tend to simplify. This paper investigates gradient descent-based learning algorithms that exhibit linearization in their parameters. We establish that this apparent linearity, arises from weak correlations between the first, and higher-order derivatives of the hypothesis function with respect to the parameters, at initialization. Our findings indicate that these weak correlations fundamentally underpin the observed linearization phenomenon of wide neural networks. Leveraging this connection, we derive bounds on the deviation from linearity during stochastic gradient descent training. To support our analysis, we introduce a novel technique for characterizing the asymptotic behavior of random tensors. We validate our theoretical insights through empirical studies, comparing the linearized dynamics to the observed correlations.

## 1 Introduction

Deep learning in general, and particularly over-parameterized neural networks, revolutionized various fields Graves et al. (2013); He et al. (2016); Krizhevsky et al. (2012); Silver et al. (2016), and they are likely to do much more. Yet, the underlying reason for their unprecedented success remains elusive. These systems can be interpreted as non-linear dynamical physical systems, characterized by a multitude of interacting degrees of freedom, which makes an exact description of their behavior exceedingly hard. However, it is well established that dynamical physical systems, when expanded to an infinite number of degrees of freedom, tend to exhibit a simplified form of dynamics Anderson (1972); therefore, it seems plausible to consider such a limit in the context of deep learning systems.

A seminal study in 2018 Jacot et al. (2018) demonstrated that wide, fully connected neural networks undergoing deterministic gradient descent behave as though they were linear with respect to their parameters (while maintaining a highly non-linear structure in their inputs). This structure has been denoted as the neural tangent kernel (NTK). The result sparked a plethora of subsequent research, generalizing it to other architectures, investigating the rate of convergence towards this linear limit, exploring the deviation of the parameters themselves from their initial configuration, decoding the structure of the kernels, and leveraging this knowledge to enhance our understanding of wide neural networks in general Lee et al. (2019); Li et al. (2019); Cao & Gu (2019); Karniadakis et al. (2021); Huang et al. (2021); Bartlett et al. (2021); Woodworth et al. (2020).

Subsequent discussions arose regarding the role of this limit in the exemplary performance of wide neural networks. Several studies have demonstrated that in certain contexts, infinitely wide neural networks converge to their global minimum at an exponential rate Jacot et al. (2018); Lee et al. (2019); Du et al. (2019); Allen-Zhu et al. (2019a;b); Daniely (2017); Li & Liang (2018); Du et al. (2018); Xu et al. (2020). Moreover, wide neural networks have been posited as effective tools for generalization, with connections drawn to the double descent phenomenon Belkin et al. (2019); Nakkiran et al. (2021); Mei & Montanari (2022). Although simplified, first-order approximations were shown to capture many of the critical properties of finite-width neural networks, making it a

valuable framework for understanding neural network behavior in general Li et al. (2019); Littwin et al. (2021); Yang & Hu (2020).

These conclusions however encounter some contention when juxtaposed with empirical evidence. Notably, several experiments indicate that for real-world data, NTK-based learning is less effective than its wide (albeit finite) neural network counterparts Lee et al. (2020); Fort et al. (2020). This apparent "*NTK inferiority paradox*" suggests that the relationship between the NTK limit and the success of finite neural networks may be more intricate than initially presumed.

A relatively understudied aspect within the framework of the neural tangent kernel pertains to the fundamental mechanisms underpinning the phenomenon of linearization. Previous research, such as Chizat et al. (2019), suggests that any gradient-based learning algorithm inherently possesses an intrinsic scale dictating its linearization behavior. Furthermore, incorporating an external parameter can modify this intrinsic scale, thereby directly influencing the extent to which linearization manifests.

In a related context, Liu et al. (2020) demonstrated that the ratio between the spectral norm of the Hessian and the Euclidean norm of the gradient governs the rate of linearization. Their analysis also established that, for wide neural networks, this ratio typically remains small, thus facilitating linearization.

Another relevant result in this field, presented by Liu et al. (2022), who proposed that the linear behavior observed in wide neural networks emerges fundamentally due to their structural composition as ensembles of numerous weak sub-models.

The closest study to this work is that of Dyer & Gur-Ari (2019), which introduced a methodology grounded in Feynman diagrams to systematically analyze wide neural networks. Their technique enables precise computation of the asymptotic behavior of correlation functions, notably the NTK, in the limit of infinite network width. By leveraging methods from theoretical physics, their work derives finite-width corrections to training dynamics, thus providing deeper insights into the evolutionary behavior of wide neural networks beyond the infinite-width approximation. However, their results are limited in scope, as their setup is restricted, and only considers the average values of these correlations.

## 1.1 OUR CONTRIBUTIONS

1. We establish that for gradient descent-based learning, linearity is equivalent to weak correlations between the first and subsequent derivatives of the hypothesis function with respect to its parameters at their initial values (Section 3.3). This equivalence is suggested as the fundamental cause for the linearization observed in wide neural networks.

2. We prove directly that wide neural networks display this weak derivative correlations structure. By relying on, and extending the tensor programs formalism Yang & Littwin (2021), our approach uniformly addresses a broader spectrum of architectures at once than any other proof we are aware of (Section 4.2).

3. Drawing from the same concepts, we demonstrate how modifications in the architecture of linearizing learning systems, and more specifically, wide neural networks, affect the linearization rate. This finding is juxtaposed with Chizat et al. (2019)'s result regarding the implications of the introduction of an external scale (Sections 3.3.2 and 4.2).

4. Harnessing weak correlations formalism, we derive a bound on the deviation from linearization over time during learning when utilizing stochastic gradient descent (Section 4.1). This is a generalization of the traditional result for deterministic gradient descent Lee et al. (2019). This is crucial, as in most practical scenarios, stochastic gradient descent generalizes better than deterministic gradient descent Lee et al. (2020); Fort et al. (2020).

5. We introduce the notion of *random tensor asymptotic behavior* as an effective analytical tool to describe the asymptotic behavior of random tensors (Section 2). Such tensors are not only integral to machine learning, but also serve a pivotal role in diverse mathematical and physical frameworks. Understanding the typical asymptotic behavior of these tensors is relevant for addressing many questions across these fields.

The overarching simplicity and broad applicability of our findings suggest that weak derivative correlations could very well be the foundational cause for the prevalent linearization attributes observed in wide neural networks, and possibly for other linearizing systems.

## 2 RANDOM TENSOR ASYMPTOTIC BEHAVIOR

Random tensors play a fundamental role in machine learning in general, and in this work in particular. In this section, we demonstrate the effectiveness of employing the stochastic big-$O$ notation of the subordinate norm to characterize the *asymptotic behavior* of a general random tensor sequence (hereinafter referred to as a random tensor). Addressing the asymptotic behavior of such tensors involves two inherent challenges: the complexity arising from their multitude of components, and the stochastic nature of these components. In this part, we define an effective way to characterize their asymptotic behavior.

Our primary norm in this work is the *Subordinate Tensor Norm*, defined as in Kreyszig (1991):

$$\|M\| = \sup \left\{ M \cdot \left( v^1 \times \ldots \times v^r \right) \mid v^1, \ldots, v^r \in S_{N_1}, \ldots, S_{N_r} \right\} . \tag{1}$$

We provide a detailed explanation of this definition and discuss its advantages in Appendix B.1.

We combine this concept with the *Stochastic Big-O Notation*, introduced in Appendix B.2, which is defined for a sequence of random tensors, denoted by $M \equiv \{M_n\}_{n=1}^{\infty}$. Henceforth, we regard $M$ as a random tensor depending on a limiting parameter $n \in \mathbb{N}^1$. This leads us to the definition of a new asymptotic upper bound for random tensors.

Denoting $\mathcal{N} = \{f : \mathbb{N} \to \mathbb{R}^{0+}\}$ as the set of all functions from $\mathbb{N}$ to the non-negative real numbers, we introduce the following definition.

**Definition 2.1** (Asymptotic Upper Bound of Random Tensors). A random tensor $M$, as defined above, is said to be asymptotically upper bounded by $f \in \mathcal{N}$ if:

$$M = O(f) , \tag{2}$$

if and only if:

$$\forall g \in \mathcal{N} \text{ s.t. } f = o(g) : \lim_{n \to \infty} P \left( \|M_n\| \le g(n) \right) = 1 . \tag{3}$$

The lower asymptotic bound, $f = \Omega(M)$, is defined analogously, but with the inequality reversed and $g = o(f)$.

As with an infinite collection of deterministic sequences, where pointwise convergence often falls short and uniform convergence is required, we require a definition of a uniform asymptotic bound when discussing an infinite number of random tensors. This concept is rigorously defined in Appendix C.1.

**Remark 2.1.** For a finite number of tensors, it can be shown directly that the uniform bound coincides with the pointwise asymptotic bound, analogously to sequence convergence.

We demonstrate in Lemma C.6 that this notation inherits many of the properties of the norm on which it is defined, including all properties of the subordinate norm, delineated in Lemma C.1. Furthermore, it satisfies several additional useful properties, outlined in Appendix C.3.

### 2.1 PROPERTIES

**Remark 2.2.** We denote $f \le g$ or $O(f) \le O(g)$ if and only if $f = O(g)$. We also denote $f < g$ or $O(f) < O(g)$ if and only if $f = O(g)$ and $f \nsim g$, where $f \sim g$ if and only if $O(f) = O(g)$, equivalently $f = O(g)$ and $g = O(f)$. It is important to note that $f < g$ may hold without requiring $f = o(g)$.

It can be readily shown that for any random tensor $M$, there exist upper and lower bounds such that $O(h_-) \le O(M) \le O(h_+)$, with $h_- \le h_+$. Furthermore, if $h_+$ and $h_-$ satisfy $h_+ \sim h_-$, then their asymptotic behavior is unique. That is, for any other pair $h'_+, h'_-$, the relation $h_+ \sim h'_+ \sim h'_- \sim h_-$

---

[1]The results are applicable not only to $\mathbb{N}$, but also to any other set endowed with a total order.

still holds (Lemma C.5). In such cases, we say that $M$ possesses an exact asymptotic behavior, denoted by $O(h_+) = O(h_-)$.

The existence of such a pair, however, is not guaranteed, as illustrated by a random variable that, for every $n \in \mathbb{N}$, has equal probability of one-half to take the value $1$ or $n$. For this variable, the optimal upper bound is $O(n)$, and the optimal lower bound is $O(1)$, yet these bounds do not share the same limiting behavior. Analogously, deterministic sequences may exhibit similar behavior, featuring multiple distinct partial limits. In the deterministic case, however, the *limsup* and *liminf* serve as the appropriate upper and lower limits, respectively. This observation leads to the question of whether an appropriate asymptotic bound exists in the random case. It turns out that it does.

**Theorem 2.1** (Definite Asymptotic Bounds for Tensors). Consider a random tensor $M$ with limiting parameter $n$ as described above. There exists $f \in \mathcal{N}$ serving as a tight asymptotic upper bound for $M$, satisfying:

$$M = O(f) \wedge \forall g \text{ s.t. } f \not\prec g : M \neq O(g) . \tag{4}$$

Furthermore, the asymptotic behavior of $f$ is unique.

***Explanation***. Although the theorem's statement is intuitive, the challenge arises from the fact that the order defined on $\mathcal{N}$ above is not total, even when considering only asymptotic behavior. For example, none of the following relations hold:

$$\sin(\pi n) < \cos(\pi n), \quad \cos(\pi n) < \sin(\pi n), \quad \sin(\pi n) \sim \cos(\pi n) . \tag{5}$$

We address this issue by employing Zorn's lemma, as demonstrated in Appendix C.2. □

Since every such random tensor $M$ admits a unique definite asymptotic bound $f$, we refer to this bound as the *random tensor's asymptotic behavior*, and write:

$$O(M) = O(f) . \tag{6}$$

# 3 WEAK CORRELATIONS AND LINEARIZATION

## 3.1 NOTATIONS FOR SUPERVISED LEARNING

### 3.1.1 GENERAL NOTATIONS

Supervised learning involves learning a *classifier*: a function $\hat{y} : X \to Y$ that maps an input set (here $X \subseteq \mathbb{R}^{d_X}$) to an output set (here $Y \subseteq \mathbb{R}^{d_Y}$), given a dataset of its values $X' \subseteq X$, denoted as the "*target function*". This is achieved by using a *hypothesis function*, in our case of the form $F : \mathbb{R}^N \to \{f : X \to Y\}$ which depends on certain parameters $\theta \in \mathbb{R}^N$ (in the case of fully connected neural networks, for example, the weights and biases). The objective of supervised learning is to find the optimal values for these parameters such that $F$ captures $\hat{y}$ best with respect to a cost function $\mathcal{C}$, assumed here convex. We use $x \in X$ to denote elements in the input set, and $i, j = 1, \ldots, d_Y$ to denote the output vector indices. The parameters $\theta$ are enumerated as $\theta_\alpha, \alpha = 1, \ldots, |\theta| = N$, and their initial values are denoted by $\theta_0 = \theta(0)$.

We work within the optimization framework of single-input-batch gradient descent-based training, which is defined such that for every learning step $s \in \mathbb{N}$:

$$\Delta^{x_s}\theta(s) = \theta(s+1) - \theta(s) = -\eta \nabla \mathcal{C}(F(\theta)(x_s), \hat{y}(x_s))|_{\theta=\theta(s)} = \\ -\eta \nabla F(\theta(s))(x_s) \mathcal{C}'(F(\theta(s))(x_s), \hat{y}(x_s)) . \tag{7}$$

Here, $\nabla_\alpha = \frac{\partial}{\partial \theta_\alpha}$ represents the gradient operator, $x_s$ denotes the $s \in \mathbb{N}$-th input data, and $\mathcal{C}'(y) = \nabla_y \mathcal{C}(y)$ refers to the derivative of the cost function. The derivative matrix/the Jacobian $\nabla F$ is defined such that for every index pair $i, \alpha$, $(\nabla F)_{\alpha i} = \nabla_\alpha F_i$. We denote $\eta$ as the learning rate and $(x_s, \hat{y}(x_s))$ as the inputs and labels, respectively. The training path is defined as the sequence of inputs upon which we trained our system, represented by $\{x_s \in X'\}_{s=0}^\infty$. We assume that each input along this path is drawn from the same random distribution $\mathcal{P}$, neglecting the possibility of drawing the same input multiple times. The same distribution is used for both training and testing. Moreover, we assume that the hypothesis function and the cost function $F, \mathcal{C}$ are analytic in their parameters. We study learning in the limit where the number of parameters $N \equiv |\theta| \to \infty$, with $N \equiv N(n)$ being a

function of some other parameter $n \in \mathbb{N}$, denoted as the "limiting parameter". For neural networks, $n$ is typically chosen as the width of the smallest layer, but we can choose any parameter that governs the system's linearization.

**Remark 3.1.** This framework can be greatly generalized, as we discussed in Appendix I.

### 3.1.2 NEURAL TANGENT KERNEL NOTATIONS

Numerous gradient descent learning systems (GDML) with different neural network architectures display a linear-like structure in their parameters in the large-width limit. In this linear limit, the hypothesis function takes the following form:

$$F_{lin}(0) = F(\theta_0), \forall s \in \mathbb{N}_0 : F_{lin}(s+1) = \\ F_{lin}(s) - \Theta_0(\cdot, x_s) \mathcal{C}'(F_{lin}(s)(x_s), \hat{y}(x_s)) \ , \tag{8}$$

with the kernel $\Theta$ defined as:

$$\forall x, x' \in X : \Theta(\theta)(x, x') = \eta \nabla F(\theta)(x)^T \nabla F(\theta)(x'), \Theta_0 \equiv \Theta(\theta_0) \ , \tag{9}$$

where $\nabla F^T$ is the transpose of $\nabla F$, the Jacobian.

## 3.2 THE DERIVATIVES CORRELATIONS

### 3.2.1 THE DERIVATIVES CORRELATIONS DEFINITION

In the following, we prove that linearization is equivalent to having weak correlations between the first and higher derivatives of the hypothesis function with respect to the initial parameters. We define the *derivative correlations* as follows:

**Definition 3.1** (Derivatives Correlations). We define the derivatives correlations of the hypothesis function for any positive integer $d \in \mathbb{N}$ and non-negative integer $D \in \mathbb{N}^0$ as:

$$\mathfrak{C}^{D,d}(\theta) = \frac{\eta^{\frac{D}{2}+d}}{D!d!} \nabla^{\times D+d} F(\theta)^T (\nabla F(\theta))^{\times d} \ , \tag{10}$$

where the higher order derivatives are defined such that for every $d \in \mathbb{N}$ and indices $i, \alpha_1 \ldots \alpha_d$, $\left(\nabla^{\times d} F\right)_{\alpha_1 \ldots \alpha_d, i} = \nabla_{\alpha_1} \cdots \nabla_{\alpha_d} F_i$.

More explicitly, presenting the inputs and indices of these tensors:

$$\mathfrak{C}^{D,d}(\theta)_{i_0, i_1 \ldots i_d}^{\alpha_{1+d} \ldots \alpha_{D+d}}(x_0, x_1 \ldots x_d) = \\ \frac{\eta^{\frac{D}{2}+d}}{D!d!} \sum_{\alpha_1 \ldots \alpha_d = 1}^{N} \nabla_{\alpha_1 \ldots \alpha_{D+d}}^{\times D+d} F_{i_0}(\theta)(x_0) \cdot (\nabla_{\alpha_1} F_{i_1}(\theta)(x_1) \cdots \nabla_{\alpha_d} F_{i_d}(\theta)(x_d)) \ . \tag{11}$$

The objects in Equation 10 are the correlation of the derivatives in the sense that $\alpha_1 \ldots \alpha_d$ can be viewed as random variables drawn from a uniform distribution of $\{1 \ldots N\}$, while $\theta$ and all other indices are fixed instances and hence deterministic. In this context, $\nabla^{\times D+d} F$ and $\nabla F \times \ldots \times \nabla F$ in Equation 10 can be viewed as random vectors of the variables $\alpha_1 \ldots \alpha_d$, and the summation in Equation 10 represents the (unnormalized) form of the "Pearson correlation" between the two random vectors. The overall coefficient of the learning rate $\eta^{\frac{D}{2}+d}$ serves as the appropriate normalization, as we will demonstrate in Appendix E and Appendix F. We will also denote: $\mathfrak{C}^d(\theta) \equiv \mathfrak{C}^{0,d}(\theta)$, $\mathfrak{C}^{D,d} \equiv \mathfrak{C}^{D,d}(\theta_0)$, $\mathfrak{C}^d \equiv \mathfrak{C}^d(\theta_0)$.

Essentially, $D + d$ represents the degree of the derivative under consideration when interacting with the first derivative, whereas $d$ specifies the number of copies of the first derivative involved in the interaction.

An example for these correlations is the $D = 0, d = 1$ correlation, the correlation of the first derivative with itself, the kernel (Equation 9):

$$\mathfrak{C}^1(\theta) = \eta \nabla F(\theta)^T \nabla F(\theta) = \Theta(\theta) \ . \tag{12}$$

The definition for the asymptotic behavior for these derivative correlations is slightly nuanced due to the many different potential combinations of distinct inputs. We rigorously define it in Appendix D.1.

In the remainder of the paper we will show how these correlations serve as an effective tool for the theoretical analysis of the linearization of wide neural networks. While this is their main purpose, they can also be used to evaluate linearization rate numerically.

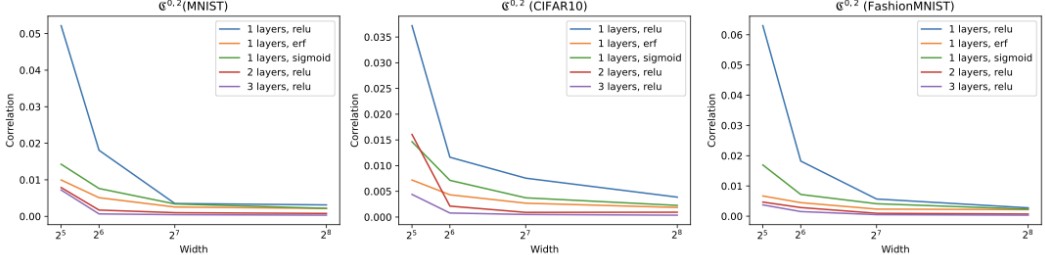

Figure 1: A comparison of the second-order correlation approximation versus the width of the network, for different datasets (MNIST, CIFAR10 and FMNIST) and for different activation functions (ReLU, Erf, Sigmoid) and numbers of layers (1,2,3).

At first glance, this may seem computationally impractical, as computing the $D, d$-th derivative requires summing $O\left(N^d\right)$ elements. However, we do not actually need to compute the full $d$-th gradient of $F$ to obtain its correlation. Instead, we can use the chain rule:

$$
\mathfrak{C}^{D,d}\left(\theta\right)_{i_0, i_1 \ldots i_d}^{\alpha_{1+d} \ldots \alpha_{D+d}}\left(x_0, x_1 \ldots x_d\right) =
$$
$$
\frac{\eta^{\frac{D}{2}+d}}{D! d!} \partial_{a_1} \cdots \partial_{a_{D+d}} F_{i_0}\left( \begin{matrix} \theta_0 + a_1 \nabla F_{i_1}\left(\theta_0, x_1\right) + \ldots + a_d \nabla F_{i_d}\left(\theta_0, x_D\right) + \\ a_{1+d} e_{\alpha_{1+d}} + \ldots + a_{D+d} e_{\alpha_{D+d}}, x_0 \end{matrix} \right). \tag{13}
$$

computed for $a_1 = \ldots = a_D = 0$, where $e_\alpha$ is the $\alpha$-th standard basis vector. This approach reduces the computation to summing only $O\left(N\right)$ elements.

### 3.3 EQUIVALENCE OF LINEARITY AND WEAK DERIVATIVES CORRELATIONS

Our main theorems concern the equivalence of linearity and weak derivative correlations. In other words, weak correlations can be regarded as the fundamental reason for the linear structure of wide neural networks. These theorems are applicable for systems that are properly scaled at the initial condition, meaning that when taking $n \to \infty$ the different components of the system remain finite. We define rigorously exactly what this means in Appendix D.2. We denote such systems as properly normalized GDMLs or *PGDMLs*.

#### 3.3.1 OUR MAIN THEOREMS

The relationship between linearization and weak derivative correlations is formalized through the equivalence theorems, which are characterized by a monotonically increasing sequence $m(n)$, where $\lim_{n \to \infty} m(n) = \infty$. This sequence captures the rate of linearization or correlation decay and constitutes an intrinsic parameter of the system. For instance, in the case of wide neural networks, one typically has $m(n) = \sqrt{n}$. Nevertheless, $m(n)$ may take any form that satisfies the stated conditions, with its mathematical role lying in defining the equivalence relation.

**Theorem 3.1** (Fixed Weak Correlations and Linearization Equivalence). Given the setup described in this section, for a sufficiently small learning rate $\eta < \eta_{the}$, the two properties are equivalent, where the asymptotic bounds are uniform for every $d, D \in \mathbb{N}$:

1. $m(n)$ - fixed weak derivatives correlation:

$$
\mathfrak{C}^d = O\left(\frac{1}{m\left(n\right)}\right), \mathfrak{C}^{D,d} = O\left(\frac{1}{\sqrt{m\left(n\right)}}\right) \tag{14}
$$

2. Simple linearity: for every fixed training step $s \in \mathbb{N}$:

$$
F\left(\theta\left(s\right)\right) - F_{lin}\left(s\right) = O\left(\frac{1}{m\left(n\right)}\right), \tag{15}
$$

$$
\eta^{\frac{D}{2}}\left(\nabla^{\times D} F\left(\theta\left(s\right)\right) - \nabla^{\times D} F\left(\theta_0\right)\right) = O\left(\frac{1}{\sqrt{m\left(n\right)}}\right). \tag{16}
$$

$\eta_{the}$ is defined such that all the correlations are uniformly bounded by $O(1)$ to ensure the sum converges, as shown in Appendix E.2. Any system that does not satisfy this condition will diverge within only a few training steps, as we show in Appendix E.2. For fully connected networks, for example, $\eta_{the} \sim \frac{1}{n}$.

The next theorem delineates an even stronger equivalence, which is also relevant for wide neural networks. It also encompasses the scaling of the learning rate.

**Theorem 3.2** (Exponential Weak Correlations and Linearization Equivalence). Given the setup described in this section, the two properties are equivalent, where the asymptotic bounds are uniform for every $D \in \mathbb{N}_0, d \in \mathbb{N}$:

1. $m(n)$ - power weak derivatives correlation: for $(D, d) \neq (0, 1)$:

$$\mathfrak{C}^{D,d} = O \left( \frac{1}{\sqrt{m(n)}} \right)^d . \tag{17}$$

2. Strong linearity: for every reparametrisation of the learning rate $\eta \to r(n)\eta, r(n) > 0$ and for every fixed training step $s \in \mathbb{N}$:

$$F(\theta(s)) - F_{lin}(s) = O \left( \frac{r(n)}{m(n)} \right) , \tag{18}$$

and for every $D \in \mathbb{N}$:

$$\left( \frac{\eta}{r(n)} \right)^{\frac{D}{2}} \left( \nabla^{\times D} F(\theta(s)) - \nabla^{\times D} F(\theta_0) \right) = O \left( \frac{r(n)}{\sqrt{m(n)}} \right) . \tag{19}$$

**Explanation:** We prove the theorems by considering, for a general learning step $s \in \mathbb{N}$, the hypothesis function and its derivatives' Taylor series expansion around step $s - 1$. Utilizing Equations 7 and 11, we find that the evolution of the derivatives of $F$ during learning is governed by a linear combination of correlations of the form:

$$\Delta \frac{\eta^{\frac{D}{2}}}{D!} \nabla^{\times D} F(\theta) = \sum_{d=1}^{\infty} \mathfrak{C}^{D,d}(\theta) \left( -\mathcal{C}'(F(\theta), \hat{y}) \right)^{\times d} , \tag{20}$$

for every $D \in \mathbb{N}_0$, where $\Delta \nabla^{\times D} F$ is the change of $\nabla^{\times D} F$. For deterministic functions, it is straightforward to prove the equivalences by employing the arithmetic properties of the big-$O$ notation, and that (i) one can choose any $F - \hat{y}$ (as long as its asymptotic behavior is appropriate), and (ii) different components in our sum cannot cancel each other, since we can change $\eta$ continuously; thus, for the sum to be small, all components must be small. The adjustments needed for our case of stochastic functions are minor, as, as we show in Appendix C.3, our tensor asymptotic behavior notation satisfies many of the same properties as the deterministic big-$O$ notation. The complete proofs are in Appendix E. These theorems describe how our weak correlations govern the transition from the "feature learning" regime to the NTK limit, as exemplified in Figure 1 and Appendix A.

### 3.3.2 RELATION TO RELATED RESULTS

As shown in Theorem 3.2, a rescaling of $\eta$, such as $\eta \to r(n)\eta$, can either promote or impede the process of linearization. This observation remains valid for Theorem 3.2 as long as $\eta < \eta_{the}$. This insight offers a deeper understanding of the findings presented by Chizat et al. (2019), specifically elucidating how an alteration of an external scale influences linearization by affecting the scales of higher-order correlations differently from those of lower-order correlations.

A notable connection to another principal work, Liu et al. (2020), concerns the definition of derivative correlations themselves. In Liu et al. (2020), the authors established that linearization results from a small ratio between the spectral norm of the Hessian and the norm of the gradient. The derivative correlations can be interpreted as a spectral norm, but concerning solely the gradient when considered as a vector. This interpretation refines the results presented in Liu et al. (2020). Unlike their approach, which required this ratio to be small within a neighborhood (ball), our framework demands its minimization specifically at the initialization point. Consequently, it necessitates the decay of higher-order correlations as well.

Another related work is the work of Huang & Yau (2020). In their work, they characterize the dynamics of wide neural networks using a hierarchy of kernels, where higher-order kernels evolve on slower time scales. Similarly to our paper, these time scales are proportional to $\frac{1}{\sqrt{n}}$, effectively capturing the deviation from the NTK limit. The relation of their work to ours is most evident in the gradient flow case, where their kernels can be expressed as linear combinations of our correlations. However, our result is more general, as it does not rely on the structural assumptions of wide neural networks, and also generalizes to finite learning rate GD. The most immediate benefit of that is that our framework applies to learning systems which are not captured by Huang & Yau (2020)'s framework. More fundamentally, by avoiding restricting ourselves only to neural networks, and instead introducing these new correlations, we obtain not only a sufficient condition for linearization, but an equivalence, providing a universal criterion applicable to generic learning systems.

The connection to Liu et al. (2022) is more abstract. Their argument, that the linear behavior observed in wide neural networks fundamentally emerges from their structural composition as ensembles of numerous weak sub-models is related to our work via the concept that neurons become independent in the infinite-width limit, precisely manifesting the absence of correlation that we emphasize.

The most closely related paper we are aware of is Dyer & Gur-Ari (2019). In this work, the authors introduced a method using Feynman diagrams to analyze wide neural networks. This approach systematically computes the asymptotic behavior of correlation functions, such as the Neural Tangent Kernel, in the large-width limit. The main difference between our and their approach is that they measure the asymptotic behavior of correlations directly, rather than averaging their values. This distinction significantly restricts their setup, rendering many of their conclusions more conjectural and less practical compared to our findings.

### 3.3.3 THE CHICKEN AND THE EGG OF LINEARIZATION AND WEAK CORRELATIONS

The relationship between linearization and weak correlations in over-parameterized systems can be comprehended from two different viewpoints. The first perspective suggests that effective learning in such systems necessitates a form of implicit regularization, which inherently favors simplicity Belkin et al. (2019). This preference can be directly incorporated by imposing a linear (or at least approximately linear) structure in highly over-parameterized regimes. Notably, in certain scenarios, linearization can facilitate exponential convergence rates, especially with respect to the training datasets, but in some instances, even with respect to the testing datasets Jacot et al. (2018); Lee et al. (2019); Du et al. (2019); Allen-Zhu et al. (2019b); Daniely (2017); Li & Liang (2018); Du et al. (2018); Xu et al. (2020); Allen-Zhu et al. (2019a). Hence, weak derivative correlations can be interpreted as a pragmatic approach for achieving linearization.

An alternative interpretation, aligning more closely with the spirit of this paper, suggests that weak derivative correlations do not primarily serve as a dynamic mechanism for linearization, but rather as its underlying cause. In this context, persisting derivative correlations may indicate an inherent bias within the system, typically an undesirable one. Therefore, linearization can be viewed as a consequence of our attempt to avoid counterproductive biases by demanding weak correlations.

This interpretation suggests that the prevalent perception of kernel learning as biased, and neural networks as unbiased, is a result-based fallacy. Had kernel learning empirically outperformed neural networks, it would seem natural to interpret linear learning in the function space (assigning large eigenvalues to simpler functions and smaller ones to complex functions) as unbiased. In other words, we interpret linear learning as overly unbiased, while finite neural networks (through mechanisms not fully understood) prioritize the inherent bias of realistic data.

Moreover, if we possess some prior knowledge about an inherent bias in our problem, it might be advantageous to allow some non-decaying correlations, counteracting the process of linearization. Furthermore, as certain biases can enhance general learning algorithms (in the form of implicit and explicit regularization), this perspective might provide valuable insights into the "NTK inferiority paradox" in the Introduction. The reason why linear learning underperforms in comparison to finite neural networks might be that it lacks some beneficial biases in the form of non-vanishing correlations.

We elaborate on this point in Appendix H.

## 4 PROPERTIES OF WEAKLY CORRELATED PGDMLS

### 4.1 APPLICATION: DEVIATION FROM LINEARITY DURING LEARNING

Multiple studies have examined the deviation of the hypothesis function $F$ from its linear approximation $F_{lin}$ (Equation 8) as a function of $n$ for a fixed learning step (especially in the context of wide neural networks). Yet, it seems that no research has explored the deviation between these functions with respect to the learning step for stochastic GD (Equation 7). This aspect is crucial since even if $F - F_{lin}$ vanishes for the initial learning steps, if it deviates too fast during learning, the linearization may not be evident for realistically large $n$.

To study how learning systems deviate from their linearization during the training process, we examine the case of an exponentially $m(n)$-weakly correlated PGDML, with learning rate satisfying $\eta < \eta_{cor}$. Here, $\eta_{cor}$ is the standard critical learning rate, ensuring that the system is stable in the NTK limit, as explained in Appendix F. We consider the problem over the span of $S \in \mathbb{N}$ learning steps, and assume that within this phase the linear solution approaches the true solution exponentially fast for some typical time $0 < T$, such that for every $s = 1 \dots S$:

$$\mathcal{C}' \left( F_{lin} \left( s \right), \hat{y} \right) = O \left( e^{-\frac{s}{T}} \right), \mathcal{C}'' \left( F_{lin} \left( s \right), \hat{y} \right) = O \left( 1 \right), \tag{21}$$

As we show in Appendix F, this is not a restrictive assumption, especially at the beginning of training, where the deviation from linearization matters the most.

**Corollary 4.1** (Weakly Correlated PGDML Deviation Over Time). Given the conditions described above, we obtain that for every $s = 1, \dots, S$:

$$F \left( \theta \left( s \right) \right) - F_{lin} \left( s \right) = O \left( \frac{s^0}{m \left( n \right)} \right). \tag{22}$$

where the asymptotic bounds are uniform in $s$, and $s^0$ denotes $s$ in the power of zero.

While this result addresses single-input-batch stochastic GD, as we explained in Appendix I, it can be greatly generalized. Notably, the analysis for stochastic GD may be more relevant even for deterministic GD than conventional approaches that presuppose a fixed training dataset. This is because, while the batch might be fixed, its initial selection is from a stochastic distribution.

***Explanation***. We prove the corollary by using a similar induction process as in Theorems 3.1and 3.2. However, here we also consider the dependency on the learning step, as detailed in Appendix F. We are able to bound the deviation over time by leveraging the fact that in the NTK limit, during the initial phases of the learning process, the system converges towards the target function exponentially fast[2] Jacot et al. (2018); Lee et al. (2019); Du et al. (2019); Allen-Zhu et al. (2019b); Daniely (2017); Li & Liang (2018); Du et al. (2018); Xu et al. (2020); Allen-Zhu et al. (2019a). We believe that subsequent research will be able to produce more refined bounds. □

### 4.2 EXAMPLE: WIDE NEURAL NETWORKS

Numerous studies have demonstrated that a wide range of neural network architectures exhibit linearization as they approach the infinite-width limit. However, existing proofs tend to be specific to particular architectures and are often intricate in nature. The most comprehensive proof we are aware of that uniformly encompasses a diverse set of architectures is presented in Yang & Littwin (2021); Yang (2020). These works employed the tensor product formalism Yang (2019), which can describe most relevant variants of wide neural network architectures as the composition of global linear operations and point-wise non-linear functions.

1. Relying on the semi-linear structure of FCNNs, we show explicitly by induction that for appropriate activation functions, wide neural networks are $\sqrt{n}$-weakly correlated, and power-weakly correlated, as shown in Appendix G.

---

[2]The known bounds for $\mathcal{C}' \left( F_{lin}, \hat{y} \right)$ are typically bounds over the variance. In Appendix C.4, we discuss how an average exponential bound can be translated into a uniform probabilistic bound.

2. The framework of low correlations proves effective in discerning how modifications to our network influence its linearization. For instance, it is evident that $\sup_{n \in \mathbb{N}} \frac{\phi^{[n]}}{(n+1)!}$ governs the rate of linearization in FCNNs (Appendix G). This observation is why we demand for FCNNs that, over the relevant domain, the activation function satisfies:

$$\phi^{[n]} \leq O\left((n+1)!\right) , \tag{23}$$

where $\phi^{[n]}$ is the $n$-th derivative of the network's activation function $\phi$.

3. Our proof for FCNNs can be generalized for any wide network described by the tensor programs formalism (Appendix G.5.1). This is because, similarly to FCNNs, all such systems exhibit a wide semi-linear form by definition. Demonstrating that the linearization of these systems arises from weak correlations allows us to utilize all of the insights we've found for weakly correlated systems in general. We also conceive linearizing network-based systems that fall outside the scope of the tensor programs formalism (Appendix G.5.2).

Leveraging the notation of the asymptotic tensor behavior, our proof accommodates a broad spectrum of initialization schemes, extending beyond the Gaussian initialization predominantly employed in other studies.

## 5 DISCUSSION AND OUTLOOK

The linearization of large and complex learning systems is a widespread phenomenon, but our comprehension of it remains limited. We propose that the weak derivatives correlations (3.1) are the underlying structure behind this phenomenon. We demonstrate that this formalism is natural for analyzing linearization: (i) it allows us to determine whether, and how fast, a general system undergoes linearization (3.3.1,4.2); and (ii) it aids us in analyzing the deviations from linearization during learning (4.1).

The strength of our approach is that it does not rely on the structure of wide neural networks. This allows us to describe not only a sufficient condition for linearization, but a true equivalence. Furthermore, it enables us to identify precisely the structural properties that generate linearization. With this, we were able to provide new insights into the linearization of wide neural networks, accounting for factors such as training duration and activation function properties. Many of these findings were previously unknown or difficult to derive using existing methods.

These insights carry practical implications. Effective systems should neither remain too close to their linear limits nor deviate excessively. A cohesive framework that relates convergence behavior to network width, training dynamics, and activation function characteristics can guide the design of more robust and efficient future models.

Our approach raises a pivotal question (3.3.3): Is the emergence of the weak correlations structure simply a tool to ensure a linear limit for overparameterized systems? Or do weak correlations indicate an absence of inherent biases, leading to linearization? If the latter is true, it suggests that in systems with pre-existing knowledge, specific non-linear learning methodologies reflecting those biases might be beneficial. That could partially explain why the NTK limit falls short in comparison to finite neural networks.

At the core of our weak derivatives correlation framework is the random tensor asymptotic behavior formalism, outlined in Section 2. We have showcased its efficacy in characterizing the asymptotic behavior of random tensors, and we anticipate its utility will extend across disciplines that involve such tensors.

We demonstrate our results empirically in Appendix A, and further discuss generalizations and limitations in Appendix I.

## ACKNOWLEDGMENTS

We would like to thank Aviv Orly for conducting the final simulations that concluded this research.

This work is supported in part by the Israeli Science Foundation excellence center, the US-Israel Binational Science Foundation, and the Israel Ministry of Science.

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

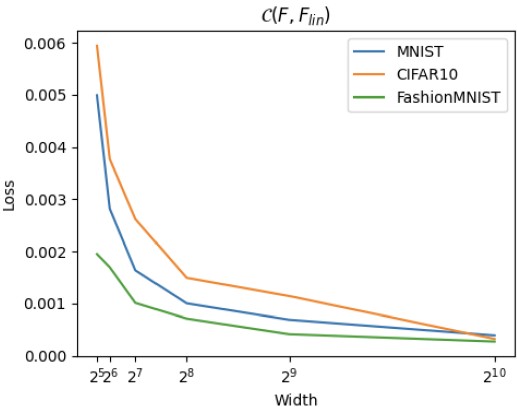

Figure 2: Relative loss between the neural network and its linear approximation versus the width, for three datasets. We used a learning rate of 1, with 1160 samples, ReLU activation, and 1000 epochs.

## A   EXPERIMENTAL RESULTS

### A.1   THEOREM 4.1 EXPERIMENTS

To support our arguments, we present numerical experiments. We show the training and testing dynamics of a neural network and its linearized approximation for varying network widths. We consider a fully connected architecture with mini-batch gradient descent, using learning rates according to the NTK normalization Lee et al. (2019), where we chose $\eta_0 = 1$. From a computational perspective, we focus on a 10-class classification task, with a total of 160 training samples and 32 test samples. We use the MSE loss for training, where each class is represented by a different one-hot vector (a 10-dimensional vector).

We perform the analysis for three datasets: CIFAR10, MNIST and FMNIST, and for different activation functions: ReLU, Sigmoid, Erf, and for different numbers of layers $L$: 1, 2, and 3 (in addition to the output layer). For instance, a network with 1 layer and width 128 on MNIST means: $784 \rightarrow 128 \rightarrow 10$.

The simulations were done in the JAX package and were based on Lee et al. (2019). We share our code on GitHub. All the results were obtained on an Apple M1 Pro CPU with 32GB of memory; the total running time was about 1 hour.

The difference function between $f$ and $f_{\text{lin}}$ was taken to be:

$$C\left(f, f_{\text{lin}}\right) = \frac{1}{10\left|X\right|}\sqrt{\sum_{x \in X}\sum_{i=1}^{10}\left(f_i(x) - f_{i,\text{lin}}(x)\right)^2}\,, \tag{24}$$

where the sub-index represents the output vector index (it depends on the class), and $X$ is the set of the data samples, which consists of 32 samples.

Calculating high-order derivatives is very costly in terms of computational resources. Therefore, we estimated the high-order partial derivatives by randomly sampling a set $D$ of weights at each layer, and averaging over a batch of samples $X$. Practically, we set $|D| = 60$ and $|X| = 160$, $d_y = 10$:

$$\mathfrak{C}^{0,2} \approx \frac{1}{S}\sqrt{\frac{1}{d_y}\sum_{i=1}^{10}\sum_{x \in X}\left(\frac{1}{\left|D\right|^2}\sum_{\alpha_1,\alpha_2 \in D}\partial_{\alpha_1}f_i\left(x\right)\partial_{\alpha_2}f_i\left(x\right)\partial_{\alpha_1}\partial_{\alpha_2}f_i\left(x\right)\right)^2}\,, \tag{25}$$

and the same goes for $\mathfrak{C}^{0,3}$.

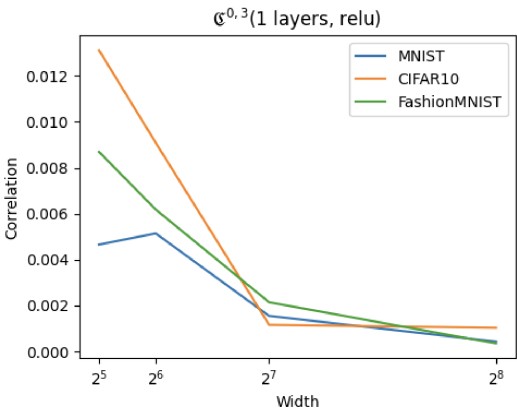

Figure 3: Third-order correlation approximation function versus different widths, for three datasets. We used a learning rate of 1, 1160 samples, ReLU activation, and 1000 epochs.

## A.2   THEOREM 3.2 EXPERIMENTS

To support our arguments, we present numerical experiments that compare the training dynamics of a nonlinear neural network with its first-order linearized (NTK/Taylor) approximation, while varying the network width and the learning-rate scaling. The goal is to quantify how accurately the linearized model tracks the true network during (finite-time) SGD training, and how the mismatch scales with width.

**Architecture and parameterization.**   We consider a fully connected MLP with two hidden layers,

$$784 \rightarrow n \rightarrow n \rightarrow 10, \tag{26}$$

using GELU activations in the hidden layers and no activation on the output logits. Importantly, we use the *standard PyTorch parameterization* (default `nn.Linear` with Kaiming/He initialization), i.e., we do *not* apply the explicit $1/\sqrt{n}$ NTK output scaling. This choice intentionally places the model outside the strict "NTK parameterization" setting and allows us to probe how learning-rate scaling alone controls the degree of "lazy" (near-initialization) training.

**Linearized model (Jacobian-based first-order approximation).**   Let $\theta_0$ denote the random initialization and $\theta$ the trainable parameters. The linearized model is the first-order Taylor expansion around $\theta_0$,

$$f_{\text{lin}}(x; \theta) = f(x; \theta_0) + J_{\theta_0}(x) (\theta - \theta_0), \tag{27}$$

where $J_{\theta_0}(x)$ is the Jacobian of $f$ with respect to parameters, evaluated at initialization. In practice, we do not form $J_{\theta_0}(x)$ explicitly. Instead, we evaluate the product $J_{\theta_0}(x) \Delta\theta$ via a Jacobian–vector product (JVP), with $\Delta\theta = \theta - \theta_0$, using PyTorch's functional API (`torch.func.functional_call`, `jvp`). The initialization $\theta_0$, the mini-batch order, and all randomness are shared between the nonlinear and linearized models for each seed.

**Dataset and task.**   We use MNIST with the standard split (60,000 training images and 10,000 test images). Images are flattened to $\mathbb{R}^{784}$ and normalized to $[0, 1]$ by division by 255, without additional standardization. Labels are converted to one-hot vectors in $\mathbb{R}^{10}$, and we train using a regression-style mean squared error (MSE) loss between logits and one-hot targets (not cross-entropy).

**Training protocol.**   We train both models using SGD (no momentum, no weight decay), batch size 128, for 250 gradient steps (about $250 \cdot 128 = 32,000$ samples, i.e., $\sim 0.53$ epochs). We chose 250 steps because, in this regime, the network behavior is stable but does not yet converge to the true solution, allowing us to isolate the effect of NTK-style linearization as cleanly as possible.

The data loader uses shuffling with a fixed seed per run and `drop_last=True` to ensure identical batch structure across models.

NTK Linearization Error vs Width by Learning Rate Scaling (α)

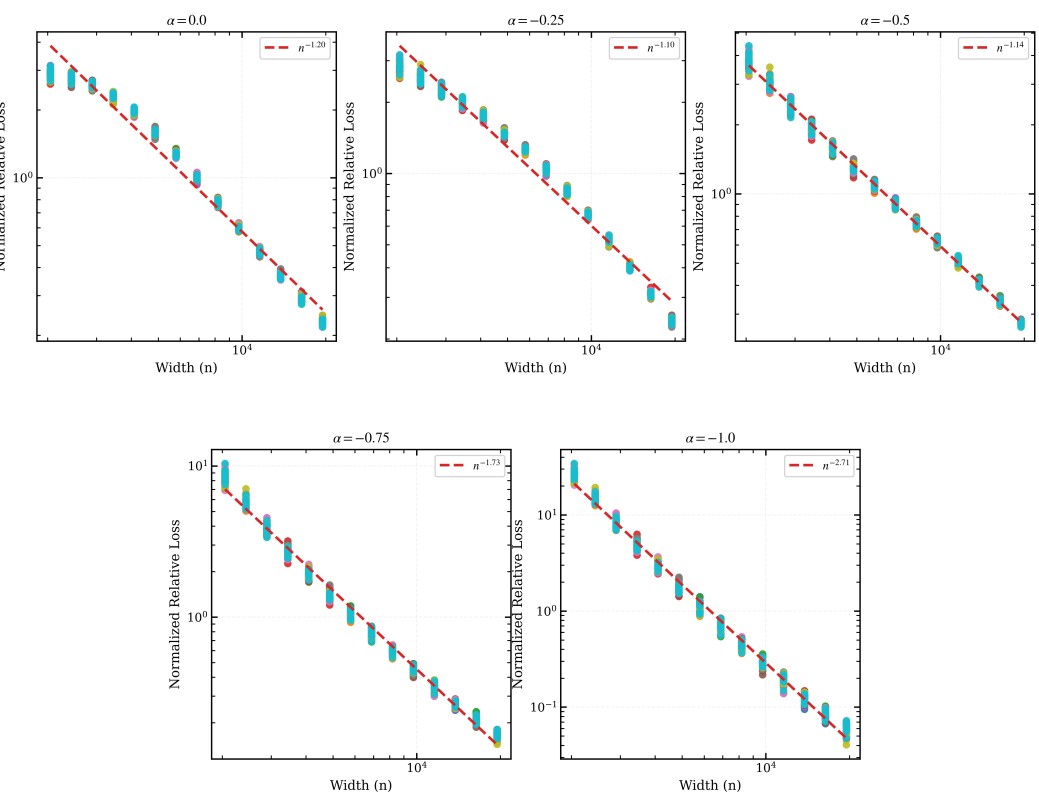

Figure 4: NTK linearization error on MNIST versus network width $n$ for five learning-rate scalings $\eta = \eta_0 n^\alpha$ (shown: $\alpha \in \{0, -0.25, -0.5, -0.75, -1\}$, with $\eta_0 = 0.1$). The y-axis reports the (normalized) test-set discrepancy between the trained nonlinear network and its first-order linearization around initialization. Each marker corresponds to an independent random seed, and the dashed line is a log–log power-law fit, yielding slopes $\approx -1.14$ ($\alpha = 0$), $\approx -1.14$ ($\alpha = -0.25$), $\approx -1.3$ ($\alpha = -0.5$), $\approx -1.73$ ($\alpha = -0.75$), and $\approx -2.72$ ($\alpha = -1$).

**Width sweep and randomness.** We sweep over 14 widths,

$$n \in \{2048, 2435, 2896, 3444, 4096, 4871, 5793, 6889, 8192, 9741, 11585, 13777, 16384, 19484\},$$
(28)

chosen to be approximately multiplicatively spaced by $2^{1/4}$ (four points per doubling). For statistical stability, we average across 100 random seeds; each seed controls both initialization and data ordering, and the nonlinear/linearized pair shares the same seed.

**Learning-rate scaling.** We use a width-dependent learning rate

$$\eta(n) \ = \ \eta_0 \, n^\alpha, \qquad \eta_0 = 0.1, \qquad \alpha \in \{0, -0.25, -0.5, -0.75, -1.0\}. \tag{29}$$

Thus $\alpha = 0$ corresponds to a constant learning rate, while $\alpha = -1$ enforces $\eta(n) = \eta_0/n$, which promotes "lazy" dynamics (smaller parameter displacement) as width grows.

**Logged metrics and key discrepancy measure.** During training (every 10 steps), we log the nonlinear and linearized training losses and the instantaneous mini-batch discrepancy between their outputs. After training, we evaluate on the full MNIST test set and define the key metric

$$\mathrm{MSE}_{\text{test}}(f, f_{\text{lin}}) \ = \ \frac{1}{|X_{\text{test}}|} \sum_{x \in X_{\text{test}}} \|f(x; \theta_{\text{nl}}) - f_{\text{lin}}(x; \theta_{\text{lin}})\|_2^2 \,, \tag{30}$$

where $|X_{\text{test}}| = 10{,}000$ and the norm is over the 10-dimensional logit vector. Lower values indicate a more faithful linearization.

**Scaling analysis.** For each $\alpha$, we fit a power law in width by linear regression in log–log coordinates,

$$\log\!\Big(\mathrm{MSE}_{\text{test}}(f, f_{\text{lin}})\Big) \;=\; s(\alpha)\,\log(n) \;+\; b(\alpha), \tag{31}$$

and report the slope $s(\alpha)$, which quantifies how rapidly the linearization error decreases with width under the learning-rate schedule $\eta(n) = \eta_0 n^\alpha$.

**Compute and reproducibility.** The experiment comprises 5 learning-rate exponents $\times$ 14 widths $\times$ 100 seeds $= 7000$ training runs. All runs were executed on an NVIDIA B200 GPU using CUDA and PyTorch (`torch.func`) for JVP-based linearization. Results are stored per-seed and per-configuration in JSON files containing the full set of hyperparameters, learning rate, step-wise losses, and the final test metrics in equation 30, enabling full reproducibility.

As shown in Figure 4, the empirical trends do not perfectly match the theoretical expectation: the observed slopes are falter than predicted. Several mechanisms may explain this gap. First, the networks may still be too small for the asymptotic regime to fully emerge. Second, our theoretical results provide upper bounds, and the two models may stay closer in practice because they learn the same data, which can mask the scaling in finite time.

## B  ADDITIONAL MATHEMATICAL BACKGROUND

In this section, we elaborate on several mathematical concepts that form the foundation for the ideas introduced in Section 2. We begin by defining the subordinate tensor norm and its key properties, then introduce a stochastic variant of the "Big O" notation to characterize the asymptotic behavior of random tensors.

### B.1  THE SUBORDINATE TENSOR NORM

Let $M$ be a tensor of rank $r \in \mathbb{N}_0$. Denote all its indices using the vector $\vec{i}$, such that each $i_e$ for $e = 1 \ldots r$ can assume values $i_e = 1 \ldots N_e$. Consequently, the tensor comprises a total of $N = N_1 \cdots N_r$ elements.

We will use the *subordinate norm*, defined as Kreyszig (1991):

$$\begin{aligned}
\|M\| &= \sup\Big\{ M \cdot \big(v^1 \times \ldots \times v^r\big)\,\Big|\, v^{1\ldots r} \in S_{N_{1\ldots r}} \Big\} = \\
&\sup\Big\{ \textstyle\sum_{i_1 \ldots i_r = 1}^{N_1 \ldots N_r} \big(M_{i_1 \ldots i_r} v^1_{i_1} \cdots v^r_{i_r}\big)\,\Big|\, v^{1\ldots r} \in S_{N_{1\ldots r}} \Big\},
\end{aligned} \tag{32}$$

where $S_{N_k} = \big\{ v \in \mathbb{R}^{N_k} : v \cdot v = 1 \big\}$ represents the unit vectors of the appropriate dimensions. This norm satisfies certain algebraic properties outlined in Lemma C.1, including: [i] the triangle inequality; [ii] for a tensor $M$ and vectors $v^1, \ldots, v^r$ with appropriately defined product, the condition $\big\| M \cdot \big(v^1 \times \ldots \times v^r\big)\big\| \leq \|M\| \|v^1\| \cdots \|v^r\|$ holds; [iii] given two tensors $M^{(1)}_{\vec{i}_1}, M^{(2)}_{\vec{i}_2}$, defining $M_{\vec{i}_1, \vec{i}_2} = M^{(1)}_{\vec{i}_1} M^{(2)}_{\vec{i}_2}$, then $\|M\| = \big\|M^{(1)}\big\| \big\|M^{(2)}\big\|$.

Also, one has $\|M\| \leq \|M\|_F$ (with equality for vectors) (Lemma C.2), where the Frobenius norm is:

$$\|M\|_F^2 = \sum_{\vec{i}} M_{\vec{i}}^2 \,. \tag{33}$$

### B.2  EFFECTIVENESS OF THE STOCHASTIC "BIG O" NOTATION

Consider a general random tensor sequence, denoted by $M \equiv \{M_n\}_{n=1}^\infty$, which henceforth we will consider as a random tensor that depends on a limiting parameter $n \in \mathbb{N}^3$.

---

[3]The results are applicable not only for $\mathbb{N}$, but also for any other set possessing a total order

Our objective in this section is to identify a method to describe and bound the asymptotic behavior of such a tensor that adheres to elementary algebraic properties. Specifically, we aim for the product of multiple bounded random tensors to be constrained by the product of their respective bounds.

Employing our defined norm (32), we can simplify our problem from general random tensors to positive random variables (rank-zero tensors), as our norm satisfies the elementary algebraic properties established in Lemma C.1. This reduction is substantial; however, the challenge of addressing the non-deterministic nature of our variable remains.

One might initially consider the expectation value of the tensor's norm as a solution. This approach unfortunately falls short, because for two positive random variables $M_1, M_2$ their product variance is not bounded by the product of their variances. In fact, for $M_1 = M_2$, the converse is true:

$$\text{Var}\left(M_1 M_2\right) \geq \text{Var}\left(M_2\right) \text{Var}\left(M_1\right) . \tag{34}$$

This issue becomes more pronounced when considering the product of multiple such variables, a frequent occurrence in this work. For instance, even with a basic zero-mean normal distribution with standard deviation $\sigma$, the higher moments of this distribution factor as $p!! = p(p-2)(p-4)\cdots$:

$$\forall p \in \mathbb{N} : \langle M^p \rangle = p!!\sigma^p . \tag{35}$$

When multiplying multiple such variables, these factors can accumulate in the lower moments, rendering this definition impractical for our purposes. Similarly, any attempt to define asymptotic behavior using the variable's moments encounters comparable difficulties.

To circumvent these challenges, we adopt the stochastic big-$O$ notation Dodge (2003); Bishop et al. (2007)[4]

## C    RANDOM TENSORS ASYMPTOTIC BEHAVIOR

In the following sections, we repeatedly use the results of this section. Due to their intuitive nature, we may not consistently specify when we do so, or which lemma/theorem we are employing.

### C.1    PROPERTIES OF OUR NORM

In this subsection, we explore the properties satisfied by the subordinate norm. We omit the proofs as these properties are either well known or straightforward to prove (and also enjoyable to derive).

**Lemma C.1** (Algebraic properties of the subordinate norm). The subordinate tensor norm (32) satisfies the following algebraic properties:

1. Given a tensor sequence $\left\{M^{(d)}\right\}_{d=1}^{D}$ where $D \in \mathbb{N} \cup \{\infty\}$, it satisfies the triangle inequality:

$$\left\|\sum_{d=1}^{D} M^{(d)}\right\| \leq \sum_{d=1}^{D} \left\|M^{(d)}\right\| , \tag{36}$$

   where equality holds when the tensors are positively linearly dependent.

2. Given a tensor $M_{i_1 \ldots i_r}$, $1 \leq i_k \leq N_k$ for $1 \leq k \leq r$, and $q \leq r \in \mathbb{N}$ vectors $v_{i_1}^1 \ldots v_{i_q}^q$ (with the same range of indices), then:

$$\left\|M \cdot v^1 \times \cdots \times v^q\right\| \leq \|M\| \left\|v^1\right\| \cdots \left\|v^q\right\| . \tag{37}$$

3. Given two tensors $M_{\vec{i}_1}^{(1)}$ and $M_{\vec{i}_2}^{(2)}$, their direct product $M_{\vec{i}_1 \vec{i}_2} = \left(M^{(1)} * M^{(2)}\right)_{\vec{i}_1 \vec{i}_2} = M_{\vec{i}_1}^{(1)} M_{\vec{i}_2}^{(2)}$, satisfies:

$$\|M\| = \left\|M^{(1)}\right\| \left\|M^{(2)}\right\| . \tag{38}$$

   The generalization to an arbitrary finite number of tensors is trivial.

---

[4]Our definition slightly differs from the standard definition of big-$O$ in probability notation, but it is straightforward to show their equivalence.

**Remark C.1.** Parts 1 and 3 are also satisfied by the Frobenius norm.

**Lemma C.2** (Relation to the Frobenius Norm). Given a tensor $M$ of rank $r \in \mathbb{N}$, the following holds:

1. For any tensor $M$:
$$\|M\| \leq \|M\|_F \ , \tag{39}$$
and if $r = 1$ (i.e., the tensor is a vector), then:
$$\|M\| = \|M\|_F = \sqrt{\sum_i M_i^2} \ . \tag{40}$$

2. For every $r' = 1 \ldots r$:
$$\|M\| = \sup \left\{ \left\| M \cdot \left( \begin{array}{c} v^1 \times \ldots \times v^{r'-1} \times \\ v^{r'+1} \times \ldots \times v^r \end{array} \right) \right\|_F \ \middle| \ \begin{array}{c} v^1 \in S_{N_1} \ldots v^{r'-1} \in S_{N_{r'-1}} \\ v^{r'+1} \in S_{N_{r'+1}} \ldots v^r \in S_{N_r} \end{array} \right\} \ . \tag{41}$$

The first part of the lemma demonstrates that our norm is always bounded by the Frobenius norm, and the two norms coincide for vectors. The second part generalizes the first, indicating that when reducing any tensor to a vector, the two norms once again agree.

**Lemma C.3** (Properties of the Maximizing Vectors). Given a tensor $M$ of rank $r \in \mathbb{N}$, there exist vectors $v^1 \ldots v^r$ of norm 1 such that:
$$\|M\| = M \cdot v^1 \times \cdots \times v^r \ . \tag{42}$$

This result indicates that the supremum is indeed a maximum. The vectors $v^1 \ldots v^{r'-1}, v^{r'+1} \ldots v^r$ are also the ones that maximize the cases demonstrated in the previous lemma.

Moreover, if the tensor is symmetric with respect to the permutation of the indices $i_1, i_2, \ldots, i_q$ and is non-zero, then:
$$v^{i_1} = v^{i_2} = \cdots = v^{i_q} \ . \tag{43}$$

**Remark C.2.** For $M = 0$, any set of vectors maximizes our result, irrespective of whether the vectors are identical or distinct.

## C.2 Existence and Uniqueness of the Tensor Asymptotic Behavior

In this section, we discuss some of the more general properties that the tensor asymptotic behavior notation satisfies, regardless of the norm it is defined with respect to. The first lemma we present is a useful equivalent definition for bounding tensor asymptotic behavior. This equivalent definition will be beneficial for our later discussion.

**Lemma C.4** (Equivalent Definitions for Tensor's Asymptotic Bound). For any random tensor $M$ and $f \in \mathcal{N}$, the two definitions for bounding the tensor's asymptotic behavior $O(M) \leq O(f)$ are equivalent (the first is the original definition, (2.1)):

1.
$$\forall g \in \mathcal{N} \text{ s.t. } f = o(g): \lim_{n \to \infty} P(\|M_n\| \leq g(n)) = 1 \ . \tag{44}$$

2.
$$\lim_{c \to \infty} \lim_{n \to \infty} P(\|M_n\| \leq cf(n)) = 1 \ . \tag{45}$$

(The same applies for $O(f) \leq O(M)$).

The order in which we take the limits in Equation 45 is crucial, as any random tensor satisfies the equation for any $f$ if we take the limit in $c$ first.

It is straightforward to show that any random tensor $M$ has lower and upper bounds:

**Lemma C.5** (Bounding Tensor Asymptotic Behavior). Given a random tensor $M$, there exist $h_-, h_+ \in \mathcal{N}$ such that:
$$O(h_-) \leq O(M) \leq O(h_+) \ . \tag{46}$$

To prove that the asymptotic tensor behavior has meaning, we need to show that bounds not only always exist, but that **there is always one well-defined "best" upper bound** (Theorem 2.1). We prove this theorem after Lemma C.5 by using Zorn's lemma.

**Remark C.3.** It is simple to show that if there exist lower and upper bounds such that $h_+ = h_-$ and the exact asymptotic behavior is well defined, then they are the "definite bound" of Theorem 2.1.

*Proof - Lemma C.4.*

We will prove the two directions of the lemma separately.

Assuming the second condition in Equation 45 is satisfied:

Given some $0 < p < 1$, we know using Equation 45 that there is some $0 < c$ such that for sufficiently large $n \in \mathbb{N}$:

$$p \leq P\left(\|M_n\| \leq cf(n)\right) . \tag{47}$$

Given some $g \in \mathcal{N}$ such that $f = o(g)$, we know that for sufficiently large $n \in \mathbb{N}$:

$$cf(n) \leq g(n) , \tag{48}$$

which means that for sufficiently large $n \in \mathbb{N}$:

$$p \leq P\left(\|M_n\| \leq cf(n)\right) \leq P\left(\|M_n\| \leq g(n)\right) . \tag{49}$$

As we proved that for any $0 < p < 1$ we get that:

$$\lim_{n \to \infty} \left(P\left(\|M_n\| \leq g(n)\right)\right) = 1 . \tag{50}$$

And as we proved that for any arbitrary $g \in \mathcal{N}$ such that $f = o(g)$, we proved the first part of the lemma.

Assuming the first condition in Equation 44 is satisfied:

If we assume in contradiction that Equation 45 is not satisfied, we get that there is some $0 < p < 1$ such that:

$$\forall n_0 \in \mathbb{N}, \, 0 < c, \, \exists n_0 \leq n \in \mathbb{N} : P\left(\|M_n\| \leq cf(n)\right) < p . \tag{51}$$

In particular, this means that if we choose the sequence $\{c_i = i\}_{i=1}^{\infty}$, there are $\tilde{n}_1 < \tilde{n}_2 < \tilde{n}_3 \ldots \in \mathbb{N}$ such that:

$$\forall i \in \mathbb{N} : P\left(\|M_{\tilde{n}_i}\| \leq if(\tilde{n}_i)\right) < p . \tag{52}$$

The reason that we can require that $\{\tilde{n}_i\}_{i=1}^{\infty}$ is increasing is that we know that we can find such $n$-values for any sufficiently large $n_0$ and for any $c$. So by induction we can require every time that every $\tilde{n}_i$ is bigger than all previous $\tilde{n}$-values.

Assuming the second condition of Equation 45 is satisfied:

Suppose, by contradiction, that Equation 44 is not satisfied. Then, there exists some $0 < p < 1$ such that:

$$\forall n_0 \in \mathbb{N}, \, 0 < c, \, \exists n_0 \leq n \in \mathbb{N} : P\left(\|M_n\| \leq cf(n)\right) < p . \tag{53}$$

In particular, if we choose the sequence $\forall i \in \mathbb{N} : c_i = i$, there exist $\tilde{n}_1 < \tilde{n}_2 < \tilde{n}_3 \cdots \in \mathbb{N}$ such that:

$$\forall i \in \mathbb{N} : P\left(\|M_{\tilde{n}_i}\| \leq if(\tilde{n}_i)\right) < p . \tag{54}$$

Since we can find such $n$-values for any sufficiently large $n_0$ and any $c$, we can require by induction that each $\tilde{n}_i$ is greater than all previous $\tilde{n}$-values.

We can now define the function:

$$\forall n \in \mathbb{N} : g(n) = \left(\max\{i \in \mathbb{N} \mid \tilde{n}_i \leq n\}\right) f(n) . \tag{55}$$

Since $\{\tilde{n}_i\}_{i=1}^{\infty}$ is increasing, we know by the Archimedean property that $\max\{i \in \mathbb{N} \mid \tilde{n}_i \leq n\}$ is also increasing and unbounded, which implies:

$$\lim_{n \to \infty} \frac{g(n)}{f(n)} = \lim_{n \to \infty} \max\{i \in \mathbb{N} \mid \tilde{n}_i \leq n\} = \infty . \tag{56}$$

However, by using Equations 52 and 55, we also have:

$$\forall n_0 \in \mathbb{N}, \, \exists n_0 \leq n \in \mathbb{N} : P\left(\|M_n\| \leq g(n)\right) < p , \tag{57}$$

which means that:

$$\lim_{n \to \infty} P\left(\|M_n\| \le g(n)\right) \ne 1 . \tag{58}$$

This contradicts our assumption in Equation 44. Therefore, by reductio ad impossibile, Equation 45 must be satisfied, completing the proof for the second direction. □

### *Proof - Lemma C.5.*

For a trivial lower bound, we choose $h_-$ such that $\forall n \in \mathbb{N} : h_-(n) = 0$.

We define $h_+$ as follows:

$$\forall n \in \mathbb{N} : h_+(n) = \inf \left\{ m \in \mathbb{R} \,\middle|\, 1 - \frac{1}{n} \le P\left(\|M_n\| \le m\right) \right\} . \tag{59}$$

The infimum and the function are well defined because:

1. The set is well defined.

2. The set is non-empty; if it were empty, it would imply that there is some probability that $\|M\|$, which is a positive number, is larger than any real number, which is impossible.

3. The set is defined with a total order " $<$ " and has a lower bound, $m = 0$.

Since for any $0 < p < 1$, there exists some $n_0 \in \mathbb{N}$ such that:

$$\forall n_0 \le n \in \mathbb{N} : p \le P\left(\|M_n\| \le m\right), \tag{60}$$

we know that for any $h_+ < g \in \mathcal{N}$, this is also true, which implies:

$$O(M) \le O(h_+), \tag{61}$$

completing the proof. □

### *Proof - Theorem 2.1.*

General Idea of the Proof:

The proof proceeds as follows:

- We consider the set of all upper bounds for $M$, denoted by $\mathcal{Z}$, and use Zorn's lemma to show that every chain[5] in this set has a lower bound within $\mathcal{Z}$.

- Applying Zorn's lemma again, we demonstrate that $\mathcal{Z}$ has a minimum.

- We then show that the limiting behavior of this minimum is unique.

Existence of an Infimum for the Upper Bound Set:

We begin by defining the set:

$$\mathcal{Z} = \left\{ h \in \mathcal{N} \,\middle|\, O(M) \le O(h) \right\} . \tag{62}$$

This set is:

1. Well defined.

2. Non-empty (as proven in Lemma C.5).

3. Defined with a partial order $h_1 < h_2 \leftrightarrow O(h_1) < O(h_2)$.

---

[5]A chain, as defined in set theory, is a subset for which the given partial order becomes a total order.

According to Zorn's lemma, if all chains in this set have a lower bound in $\mathcal{Z}$, then $\mathcal{Z}$ has at least one minimum.

Given some chain in the set, $\mathcal{C} \subseteq \mathcal{Z}$, we know it is lower bounded by the function $h_-$, which means (by using Zorn's lemma) it has at least one infimum (a lower bound without any larger lower bounds). We choose such an infimum and denote it by $I \in \mathcal{N}$.

Proving that the Infimum is in $\mathcal{Z}$:

We assume, by contradiction, that this infimum is not in $\mathcal{Z}$, which means there exists some $g \in \mathcal{N}$ such that $I = o(g)$ and for every $0 < p < 1$ and $n_0 \in \mathbb{N}$, there exists $n_0 \leq n \in \mathbb{N}$ such that:

$$P\left(\|M_n\| \leq g\left(n\right)\right) < p \, . \tag{63}$$

Since $I = o(g)$, we know that for any $c \in \mathbb{R}$ and sufficiently large $n \in \mathbb{N}$:

$$cI\left(n\right) \leq g\left(n\right) \, . \tag{64}$$

Combining these equations, we obtain:

$$\forall 0 < c, \, n_0 \in \mathbb{N} \, \exists n_0 \leq n \in \mathbb{N} : P\left(\|M_n\| \leq cI\left(n\right)\right) < p \, . \tag{65}$$

In particular, if we choose the sequence $\forall i \in \mathbb{N} : c_i = i^2$, there exist $\tilde{n}_1 < \tilde{n}_2 < \tilde{n}_3 \ldots \in \mathbb{N}$ such that:

$$\forall i \in \mathbb{N} : P\left(\|M_{\tilde{n}_i}\| \leq i^2 I\left(\tilde{n}_i\right)\right) < p \, . \tag{66}$$

We can require that $\{\tilde{n}_i\}_{i=1}^{\infty}$ is increasing for the same reason as before, as we know that we can find such arbitrarily large $n$-values for any sufficiently large $n_0$ and for any $c$, so we can, by induction, demand that each $\tilde{n}_i$ is greater than all previous $\tilde{n}_1 \ldots \tilde{n}_{i-1}$.

Now, we define the function:

$$J\left(n\right) = \left\{ \begin{array}{c} iI\left(n\right) : \exists i \in \mathbb{N} : n = \tilde{n}_i \\ I\left(n\right) : \text{else} \end{array} \right. \, . \tag{67}$$

This function is well defined because there is only one $i$ for any $n$ such that $n = \tilde{n}_i$, as it is an increasing sequence.

Using Equations 66 and 67, we find that the subsequence $\{\tilde{n}_i\}_{i=1}^{\infty}$ satisfies:

$$\forall i \in \mathbb{N} : P\left(\left\|M\left(q\right)_{\tilde{n}_i}\right\| \leq iJ\left(\tilde{n}_i\right)\right) < p \, . \tag{68}$$

Applying Lemma C.4 for the equivalency of the asymptotic bound definition, we conclude that above this subsequence $J \notin \mathcal{Z}$, which implies that above this subsequence $J$ is a lower bound of $\mathcal{Z}$ and consequently also of $\mathcal{C}$. Moreover, for all other $n$, we have $J = I$, and since $I$ is a lower bound of $\mathcal{C}$, so is $J$. Since every $n \in \mathbb{N}$ belongs to one of these subsequences, we conclude that $J$ is a lower bound of $\mathcal{C}$ in general.

Furthermore, for every $i \in \mathbb{N}$, as $1 \leq c_i$, we have:

$$\forall n : I(n) \leq J(n) \rightarrow O(I) \leq O(J) \, . \tag{69}$$

However, since $\{\tilde{n}_i\}_{i=1}^{\infty}$ is increasing and unbounded, we know that there exists at least one subsequence such that:

$$\lim_{\tilde{n}_i \to \infty} \frac{J\left(\tilde{n}_i\right)}{I\left(\tilde{n}_i\right)} = \lim_{i \to \infty} c_i = \infty \rightarrow O\left(J\right) \neq O\left(I\right) \, . \tag{70}$$

This implies:

$$O\left(I\right) < O\left(J\right) \rightarrow I < J \, . \tag{71}$$

We have discovered that $J$ is greater than $I$, but smaller than all functions in $\mathcal{C}$, which implies that it is a larger lower bound than the infimum, which is impossible, and implies by reductio ad impossibile that every chain in $\mathcal{Z}$ has a lower bound in $\mathcal{Z}$.

Existence and Uniqueness of the Minimum:

Using Zorn's lemma, we now know that $\mathcal{Z}$ has at least one minimum, denoted by $f \in \mathcal{N}$. Our remaining task is to show that all other minima in $\mathcal{Z}$ exhibit the same limiting behavior as $f$, which implies the uniqueness of the minimal limiting behavior.

Let $g \in \mathcal{N}$ be another minimum. We define:

$$\forall n \in \mathbb{N} : h(n) = \min\{f(n), g(n)\} . \tag{72}$$

We know that $h \leq f, g$, and we also know that $h \in \mathcal{Z}$ since $f, g \in \mathcal{Z}$ and for every $0 < p < 1$ we can choose the maximal $n_0$ from $f$ and $g$. Thus, $h \in \mathcal{Z}$, but $h \leq f, g$ as well, where $f, g$ are minima themselves. This implies:

$$O(f) = O(h) = O(g) \rightarrow O(f) = O(g) . \tag{73}$$

Therefore, there exists a unique minimal limiting behavior, which implies that the tensor's asymptotic behavior is always well defined. $\qquad\square$

**Remark C.4.** In our proof, we employed Zorn's lemma twice. First, we used it to demonstrate the existence of an infimum for every chain, and then, after showing that these infima belong to $\mathcal{Z}$, we employed it again to establish that $\mathcal{Z}$ has a minimum. At first glance, it may seem perplexing that we needed to rely on Zorn's lemma, an incredibly abstract and powerful tool equivalent to the somewhat controversial axiom of choice, to prove that the tensor's asymptotic behavior, which has a much more grounded and intuitive meaning, is well defined.

One possible explanation for this discrepancy is that we may not have actually required the full power of the axiom of choice, and our structures could be simple enough that an alternative approach could have been taken to prove our theorem without using Zorn's lemma. We believe, however, that in the most general case, Zorn's lemma was indeed necessary, but it was only relevant for extreme distributions lacking any tangible "physical meaning." For any well-defined set of distributions with a clear underlying meaning, one could potentially find an alternative method for demonstrating the existence of a tight bound without invoking Zorn's lemma.

In any case, as we demonstrated in Lemma C.5, there is no need for any of these high-level tools to prove the existence of an upper bound.

### C.3 PROPERTIES OF THE ASYMPTOTIC BEHAVIOR NOTATION

Having established that our notation is meaningful, we now aim to demonstrate its usefulness. First, we need to address our earlier issue and define a "uniform asymptotic bound." Once again, we omit the proofs in this and the following sections.

**Definition C.1** (Uniform Tensor Asymptotic Bound)**.** Given a sequence of random tensors $\left\{M^{(d)}\right\}_{d=1}^{D}$, where $D \in \mathbb{N} \cup \{\infty\}$ (or, more precisely, a sequence of random tensor sequences) with a limiting parameter $n$, we say that it is uniformly asymptotically upper bounded by $f \in \mathcal{N}$ under some rising monotonic function $\mathcal{K}^{1 \ldots D} : \mathbb{R} \to \mathbb{R}$:

$$\forall d = 1 \ldots D : O\left(M^{(d)}\right) \leq O\left(\mathcal{K}^d \circ f\right) \quad \text{Uniformly} , \tag{74}$$

if and only if:

$$\forall g \in \mathcal{N} \text{ s.t. } f = o(g) : \lim_{n \to \infty} P\left(\forall d = 1 \ldots D : \left\|M_n^{(d)}\right\| \leq \mathcal{K}^d \circ g(n)\right) = 1 . \tag{75}$$

The definition for a uniform lower asymptotic bound is analogous with reversed directions.

**Remark C.5.** As discussed in Definition 2.1, it is clear that if $D$ is finite, then a uniform bound is equivalent to a pointwise bound.

**Lemma C.6** (Asymptotic Notation Inherits its Norm Properties)**.** Given a random tensor $M$ and a sequence of jointly distributed random tensors $\left\{M^{(d)}\right\}_{d=1}^{D}$ (with $M$ as well), where $D \in \mathbb{N} \cup \{\infty\}$, such that they are all uniformly bounded:

$$\forall d = 1 \ldots D : O\left(M^{(d)}\right) \leq O\left(\mathcal{K}^d \circ f\right) \quad \text{Uniformly} , \tag{76}$$

then:

1. If some positive linear combination of $M^{(d)}$'s norms satisfies an inequality of the form:

$$\|M\| \leq \sum_{\tilde{d}=1}^{\tilde{D}} \lambda_{\tilde{d}} \prod_{d=D_{\tilde{d}-1}+1}^{D_{\tilde{d}}} \|M_d\| , \tag{77}$$

where all of the coefficients are positive: $\forall \tilde{d} = 1 \dots \tilde{D} : 0 \leq \lambda_{\tilde{d}}$ and we divided $1 \dots D$ into a sequence of finite intervals: $0 = D_1 < D_2 < \dots < D_{\tilde{D}} = D$. Then the asymptotic behavior of all the tensors satisfies the same inequality as well for every $h \sim f$:

$$O\left(M\right) \leq O\left(\sum_{\tilde{d}=1}^{\tilde{D}} \lambda_{\tilde{d}} \prod_{d=D_{\tilde{d}-1}+1}^{D_{\tilde{d}}} \mathcal{K}^d \circ h\right), \tag{78}$$

and if the inequality is an equality for the norm, it is also an equality for the "large $O$" notation.

2. Our asymptotic notation inherits all of the properties presented in Lemma C.1.

**Remark C.6.** The lemma still holds even if the tensor has additional indices, as we will see in Section G.4, provided the number of additional index possibilities remains finite in $n$.

## C.4 EXPLORING THE RELATIONSHIP BETWEEN ASYMPTOTIC BEHAVIOR NOTATION AND THE TENSORS' MOMENTS

The final aspect of the asymptotic behavior notation we wish to explore is the relationship between this notation and the moments of our tensors' norm or variables. This relationship is relatively intuitive and straightforward, and will be useful in Section G. We first introduce a simple notation for a tensor $M_{\vec{i}}$ that will assist in examining tensor moments, the norm expectation value, defined as:

$$[M] = \sqrt{\frac{1}{N} \left\langle \|M\|^2 \right\rangle}, \tag{79}$$

**Lemma C.7** (Asymptotic Behavior and Tensor Moments Equivalency). Given a random tensor $M$ and a function $f \in \mathcal{N}$, then:

$$O(M) \leq O(f), \tag{80}$$

if and only if, with probability arbitrarily close to 1:

$$[M] = O(f). \tag{81}$$

The lemma is also applicable for the uniform bound in the case of an infinite collection of random tensors.

In Section 4.1, we highlighted that most assertions concerning the convergence of $\mathcal{C}'(F - \hat{y})$ relate to its expected value. However, we can now also associate it with its asymptotic behavior throughout the entire training trajectory. This association stems from the understanding that, if our system exhibits a known average decay, the likelihood of significant deviations from this typical variance range must also decrease, and exponentially (at any decaying rate that is slower than our original rate). Given that decaying geometric sums are convergent, we can infer that the overall probability of the system defying our predicted asymptotic behavior is likewise convergent. Given that we can choose the scaling of this probability arbitrarily, we can set conditions such that the cumulative probability of any deviation is arbitrarily small. We introduce this notion for the reader's consideration and propose a detailed formulation as a future exercise.

## D ADDITIONAL DEFINITIONS

### D.1 DERIVATIVES CORRELATIONS ASYMPTOTIC BEHAVIOR

In our main text (3.2.1), we discussed that the definition for the asymptotic behavior of the derivatives correlations is slightly nuanced due to the many different potential combinations of distinct inputs. Here we define it rigorously.

**Definition D.1** (Derivatives Correlations Asymptotic Behavior). For every $D \in \mathbb{N}^0$, $d \in \mathbb{N}$, and $d_1 \leq d_2 \leq \dots \leq d_{\tilde{d}} \in \mathbb{N}$ such that $d_1 + \dots + d_{\tilde{d}} = d$:

$$O_{d_1 \dots d_{\tilde{d}}}\left(\mathfrak{C}^{D,d}\right) \equiv O_{x_0, x_1 \dots x_{\tilde{d}} \in \mathcal{P}}\left(\mathfrak{C}^{D,d}\left(x_0, x_1^{\times d_1} \dots x_{\tilde{d}}^{\times d_{\tilde{d}}}\right)\right). \tag{82}$$

The input order does not matter, as the correlations are symmetric with respect to their first derivatives. The factor $\frac{d!}{d_1! \cdots d_{\bar{d}}!}$ accounts for the possible combinations. If $f \in \mathcal{N}$, we say:

$$\mathfrak{C}^{D,d} = O(f) , \tag{83}$$

if and only if all combinations are uniformly bounded by $f$. In the continuous limit (extended training time), only $d_1 = \ldots = d_d = 1$ remains relevant.

## D.2 PROPERLY NORMALIZED GDML

Our main Theorems (3.1,3.2) and Corollary (4.1) are applicable for systems that are properly scaled at the initial condition as $n \to \infty$, defined as follows.

**Definition D.2** (PGDML).  Given a GDML as described in Section 3.1, we say that it is properly normalized (and denote it as a PGDML) if and only if:

$$F(\theta_0) = O(n^0) \tag{84}$$

$$\Delta F(\theta_0) = F(\theta(1)) - F(\theta_0) = O(n^0) \tag{85}$$

$$\mathfrak{C}^1 = (N\eta) O(\nabla F(\theta_0))^2 \tag{86}$$

$$\forall d \in \mathbb{N} : O\left(\nabla^{\times d} F(\theta_0)\right) \leq O\left(\nabla F(\theta_0)\right)^d \quad \text{Uniformly.} \tag{87}$$

Where $n^0$ symbolizes $n$ in the power of zero.

The first two conditions (84,85) ensure that our system scale remains finite at the initial condition. Condition 86 stipulates that the asymptotic behavior of the kernel is maximal, given the asymptotic behavior of the first derivative. This condition ensures that our system is genuinely learning and not only memorizing. This is because the kernel for different inputs is responsible for extrapolation, while the kernel with the same input twice is responsible for memorization[6]. Condition 87 asserts that none of the higher derivatives dominate the first for $n \to \infty$, a property that most realistic scalable GDMLs satisfy, because if it is not satisfied, gradient descent becomes irrelevant. We show that wide neural networks in general satisfy this property in Appendix G.5.

## E    PROOF OF THEOREMS 3.1,3.2

We can now proceed with the proofs of Theorems 3.1 and 3.2. The general idea has been outlined at the end of Section 3.3.1.

### E.1    FIRST DIRECTION OF THEOREMS 3.1,3.2

Now that we understand how to work with the asymptotic behavior of random tensors, we can proceed to prove our main theorems and corollary. We begin with the first direction of the theorems.

**Lemma E.1** (Linearization Requires Weak Correlations)**.**

1.  In Theorem 3.1, if condition 1 is satisfied, then condition 2 is satisfied as well.

2.  In Theorem 3.2, if condition 1 is satisfied, then condition 2 is satisfied as well.

***Proof.*** We only demonstrate that the $O_1(\mathfrak{C})$ are bounded; the proof that the rest are bounded is the same, by considering more learning steps after the initial condition.

---

[6]This is a direct consequence of the NTK equation of motion (8). For example, in the case of a single input point, the system behaves like a memorization algorithm for that one input. However, the term $\Theta(x, x')$ governs how the value of the function at $x$ is influenced by its values at other points $x'$.

For the initial condition, we know that any reparameterization $0 < r$ satisfies (8,20):

$$F\left(\theta\left(1\right)\right) - F_{lin}\left(1\right) =$$
$$\sum_{d=1}^{\infty} \frac{(r\eta)^d}{d!} \left(\nabla^{\times d} F\left(\theta_0\right)\left(\nabla F\left(\theta_0\right)\left(x_1\right)^T\right)^{\times d}\right)\left(-\mathcal{C}'\left(F\left(\theta_0\right)\left(x_1\right),\hat{y}\left(x_1\right)\right)\right)^{\times d} -$$
$$\left(-\left(r\eta\right)\nabla F\left(\theta_0\right)\nabla F\left(\theta_0\right)\left(x_1\right)^T\left(-\mathcal{C}'\left(F\left(\theta_0\right)\left(x_1\right),\hat{y}\left(x_1\right)\right)\right)\right) = \tag{88}$$
$$\sum_{d=2}^{\infty} r^d \left(\frac{\eta^d}{d!}\nabla^{\times d} F\left(\theta_0\right)\left(\nabla F\left(\theta_0\right)\left(x_1\right)^T\right)^{\times d}\right)\left(-\mathcal{C}'\left(F\left(\theta_0\right)\left(x_1\right),\hat{y}\left(x_1\right)\right)\right)^{\times d} =$$
$$\sum_{d=2}^{\infty} r^d \left(\mathfrak{C}^d\right)^{\cdot,x_1^{\times d}}\left(-\mathcal{C}'\left(F\left(\theta_0\right)\left(x_1\right),\hat{y}\left(x_1\right)\right)\right)^{\times d} ,$$

and in the same way, for every $D \in \mathbb{N}$:

$$\frac{(r\eta)^{\frac{D}{2}}}{D!}\nabla^{\times D} F\left(\theta\left(1\right)\right) - \frac{(r\eta)^{\frac{D}{2}}}{D!}\nabla^{\times D} F_{lin}\left(\theta\left(1\right)\right) =$$
$$\sum_{d=1}^{\infty} r^{\frac{D}{2}+d}\left(\mathfrak{C}^{D,d}\right)^{\cdot,x_1^{\times d}}\left(-\mathcal{C}'\left(F\left(\theta_0\right)\left(x_1\right),\hat{y}\left(x_1\right)\right)\right)^{\times d} . \tag{89}$$

Utilizing Lemma C.6, it becomes evident that for properly normalized gradient descent-based systems:

$$O\left(\mathfrak{C}^{D,d}\mathcal{C}'\left(F\left(\theta_0\right),\hat{y}\right)^{\times d}\right) \leq O\left(\mathfrak{C}^{D,d}\right) O\left(\mathcal{C}'\left(F\left(\theta_0\right),\hat{y}\right)^{\times d}\right) = O\left(\mathfrak{C}^{D,d}\right) . \tag{90}$$

However, since our theorem should work for any $\hat{y}$, we can choose $\hat{y} = F\left(\theta_0\right) + c$, and obtain:

$$O\left(\mathfrak{C}^{D,d}\mathcal{C}'\left(F\left(\theta_0\right),\hat{y}\right)^{\times d}\right) \propto O\left(\mathfrak{C}^{D,d}\mathcal{C}'\left(c\right)^{\times d}\right) = O\left(\mathfrak{C}^{D,d}\right) , \tag{91}$$

as we can choose $c$ such that $\mathcal{C}'(c)$ is the vector that maximizes the correlation, since $\mathcal{C}'$ is convex and the correlations are symmetrical.

Given that we can choose an open set of different scalings of $r$, we know the different elements in the series cannot cancel each other out. Consequently, for $F - F_{lin}$ to decay, all the distinct elements must decay.

Assuming condition 1 in Theorem 3.1:

Given that $O\left(F\left(\theta\left(1\right)\right) - F_{lin}\left(1\right)\right) = O\left(\frac{1}{m(n)}\right)$ and for every $D \in \mathbb{N}$ we have $O\left(\eta^{\frac{D}{2}}\nabla^{\times D} F\left(\theta\left(1\right)\right) - \eta^{\frac{D}{2}}\nabla^{\times D} F\left(\theta_0\right)\right) = O\left(\frac{1}{\sqrt{m(n)}}\right)$, it follows that each correlation must decay at least like:

$$\forall 2 \leq d \in \mathbb{N} : O\left(\mathfrak{C}^d\right) \leq O\left(\frac{1}{m\left(n\right)}\right) \quad \text{Uniformly,} \tag{92}$$

and

$$\forall D, d \in \mathbb{N} : O\left(\mathfrak{C}^{D,d}\right) \leq O\left(\frac{1}{\sqrt{m\left(n\right)}}\right) \quad \text{Uniformly.} \tag{93}$$

This completes the first part of the proof.

Assuming condition 1 in Theorem 3.2:

By taking $r(n)$ arbitrarily close to $m(n)$, we find that for $F\left(\theta\left(1\right)\right) - F_{lin}\left(1\right)$ to decay, $r^d\mathfrak{C}^d$ must decay as well, which implies that:

$$\forall d \in \mathbb{N} : O\left(\mathfrak{C}^d\right) \leq O\left(\frac{1}{m\left(n\right)}\right)^d , \tag{94}$$

and

$$\forall D \in \mathbb{N}^0, d \in \mathbb{N} : O\left(\mathfrak{C}^{D,d}\right) \leq O\left(\frac{1}{\sqrt{m\left(n\right)}}\right)^d . \tag{95}$$

This concludes our proof. $\qquad\square$

### E.2 SECOND DIRECTION OF THEOREMS 3.1, 3.2

We now prove the other direction of the theorems, focusing on Theorem 3.1 since the proofs for the other theorems are essentially the same. It should also be noted that Corollary 4.1, which will be proven next, is almost a generalization of this direction, except that it is only applicable for sufficiently small learning rates.

**Lemma E.2** (Asymptotic Behavior Normalization for Weakly Correlated PGDML). Consider a weakly correlated PGDML as described in Theorems 3.1and 3.2. Then we have:

$$\forall D \in \mathbb{N} : \eta^D O \left(\nabla^{\times D} F\left(\theta_0\right)\right)^2 \leq O\left(1\right) \quad \text{Uniformly.} \tag{96}$$

With Lemma E.2 at hand, we can now demonstrate the second direction of the theorem by proving a slightly stronger version of it.

**Lemma E.3** (Weak Correlations Create Linearization (First Theorem)). Assuming the conditions of Theorem 3.1 part 1, then for every $s = 1 \ldots S$:

1.
$$O\left(F\left(\theta\left(s\right)\right) - F_{lin}\left(s\right)\right) \leq O\left(\frac{1}{m\left(n\right)}\right) . \tag{97}$$

2.
$$O\left(\eta^{\frac{1}{2}}\nabla F\left(\theta\left(s\right)\right) - \eta^{\frac{1}{2}}\nabla F\left(\theta_0\right)\right) \leq \gamma . \tag{98}$$

3. For every $2 \leq D \in \mathbb{N}$

$$O\left(\eta^{\frac{D}{2}}\nabla^{\times D} F\left(\theta\left(s\right)\right) - \eta^{\frac{D}{2}}\nabla^{\times D} F\left(\theta_0\right)\right) \leq O\left(\frac{1}{\sqrt{m\left(n\right)}}\right) \quad \text{uniformly.} \tag{99}$$

Here, $\gamma$ is an asymptotic notation such that $\gamma = O\left(\frac{1}{\sqrt{m(n)}}\right)$, and when multiplied with a first derivative of the hypothesis function in its initial condition, it exhibits an asymptotic behavior of $O\left(\gamma_t \eta^{\frac{1}{2}}\nabla F\left(\theta_0\right)\right) \leq O\left(\frac{1}{m(n)}\right)$.

From proving Lemmas E.1and E.3, we can conclude that Theorems 3.1and 3.2 have been proven.

***Proof of Lemma E.2.***

Assume that the lemma is not satisfied, i.e.,

$$\eta O\left(\nabla F\left(\theta_0\right)\right)^2 \not\leq O\left(1\right) , \tag{100}$$

then for some probability $0 < p < 1$, we have:

$$O\left(1\right) < \eta O\left(\nabla F\left(\theta_0\right)\right)^2 . \tag{101}$$

Utilizing the third property of PGDML systems (86), we conclude that for some relevant probability:

$$O\left(1\right) < O\left(\mathfrak{C}^1\right) . \tag{102}$$

However, for the reasons discussed earlier, the different elements in the equation of motion cannot cancel each other out, as $\eta$ can be chosen from an open set. This implies that the second property of PGDML systems (85) cannot be satisfied, leading to the conclusion that:

$$\eta O\left(\nabla F\left(\theta_0\right)\right)^2 \leq O\left(1\right) , \tag{103}$$

must hold.

By employing the fourth property (87) of PGDML systems, we obtain the desired result. □

*Proof of Lemma E.3.*

We will prove the lemma using induction over the learning steps. The induction base for the "zero" step, where $\theta = \theta_0$, is trivial. Assuming the lemma holds for $s \in \mathbb{N}^0$, we observe that for every $\left(D \in \mathbb{N}^0, d \in \mathbb{N}\right) \neq (0, 1)$, the $(D, d)$-correlation satisfies the following for sufficiently small learning rate $\eta$:

$$
\begin{aligned}
\mathfrak{C}^{D,d}\left(\theta\left(s\right)\right) &= \eta^{\frac{D}{2}+d}\nabla^{\times D+d}F\left(\theta\left(s\right)\right)^T \nabla F\left(\theta\left(s\right)\right)^{\times d} \\
&= \\
&\left(\eta^{\frac{D+d}{2}}\nabla^{\times D+d}F\left(\theta_0\right)+\gamma\right)^T \left(\eta^{\frac{1}{2}}\nabla F\left(\theta_0\right)+\gamma\right)^{\times d} \\
&= \\
&\mathfrak{C}^{D,d} + \gamma^T\left(\eta^{\frac{1}{2}}\nabla F\left(\theta_0\right)\right)^{\times d} + \gamma^T\left(\gamma\times\left(\eta^{\frac{1}{2}}\nabla F\left(\theta_0\right)\right)^{\times d-1}\right) + \\
&\eta^{\frac{D+d}{2}}\nabla^{\times D+d}F\left(\theta_0\right)\left(\gamma\times\left(\eta^{\frac{1}{2}}\nabla F\left(\theta_0\right)\right)^{\times d-1}\right) + \text{comb} + O\left(\tfrac{1}{m(n)}\right) \\
&= \\
&\mathfrak{C}^{D,d} + O\left(\tfrac{1}{m(n)}\right) + O\left(\tfrac{1}{m(n)}\right) + d\mathfrak{C}^{D+1,d-1}\times\gamma + O\left(\tfrac{1}{m(n)}\right) \\
&= \\
&\mathfrak{C}^{D,d} + O\left(\tfrac{1}{m(n)}\right) .
\end{aligned}
\tag{104}
$$

Here, we used the derivatives correlation definition, the relevant lemmas, the induction hypothesis, the bound of the correlations from condition 1, and the definition of $\gamma$.

By employing the derivatives correlation definition and condition 1, we observe that:

$$
\begin{aligned}
\forall 2 \leq d \in \mathbb{N} : O\left(\mathfrak{C}^d\right) &= O\left(\tfrac{1}{m(n)}\right) , \\
\forall d \in \mathbb{N} : O\left(\mathfrak{C}^{1,d}\right) &= \gamma , \\
\forall 2 \leq D \in \mathbb{N}, d \in \mathbb{N} : O\left(\mathfrak{C}^{D,d}\right) &= O\left(\tfrac{1}{\sqrt{m(n)}}\right) .
\end{aligned}
\tag{105}
$$

Furthermore:

$$
\begin{aligned}
\mathfrak{C}^{1,d}\eta^{\frac{1}{2}}\nabla F\left(\theta_0\right) &= \eta^{\frac{1}{2}+d}\nabla^{\times d+1}F\left(\theta_0\right)^T\left(\eta^{\frac{1}{2}}\nabla F\left(\theta_0\right)\right)^{\times d} = \\
&\eta^{d+1}\nabla^{\times d+1}F\left(\theta_0\right)^T\left(\nabla F\left(\theta_0\right)\right)^{\times d+1} = \mathfrak{C}^{d+1} .
\end{aligned}
\tag{106}
$$

Hence, using this equation, we can deduce that $\mathfrak{C}^{D,d}\left(\theta\left(s+1\right)\right)$ satisfies the given conditions as well. By incorporating this equation into our equation of motion and employing the lemmas, we find that for a sufficiently small learning rate, $F\left(\theta\left(s+1\right)\right)$ also satisfies the lemma. Consequently, by induction, the lemma holds for all $s \in \mathbb{N}$. $\qquad\square$

# F PROOF OF COROLLARY 4.1

In this section, we prove Corollary 4.1. The general approach for this proof is akin to that of the first direction of Theorems 3.1 and 3.2, albeit with an additional focus on the evolution of the deviation throughout the induction process.

Given the complexity of tracking all the derivatives simultaneously, our strategy involves monitoring the difference between the parameters and their linearization, as expressed in Equation 113. A significant challenge arises in solving the equation of motion that these parameters must satisfy.

To circumvent this issue, we establish a link between this deviation and the deviation of the generalization function from its linearization (113) up to the highest order, as outlined in Equation 118. By considering only the lowest-order terms, we obtain an equation of motion (125). In cases where the cost function decays exponentially, we are able to bound the deviation of this equation.

## F.1 RELATIONS BETWEEN DIFFERENT LINEARIZATIONS

In the main text, we linearized $F$ as $F_{lin}$ (8), by first considering only the linear part of $F$, and then examining how it changes over time for a given training path. However, there are alternative ways to

linearize $F$ that can be useful to consider. One such method involves taking only the linear part of $F$, without considering the training path:

$$\hat{F}(\theta) = F(\theta_0) + \nabla F(\theta_0)^T (\theta - \theta_0) . \tag{107}$$

Another useful definition is to examine how $\theta$ would develop over time under the linear approximation for our training path:

$$\theta_{lin}(0) = \theta_0 \quad \forall s \in \mathbb{N}:$$
$$\theta_{lin}(s+1) = \theta_{lin}(s) - \nabla F(\theta_0)(x_s) C'(F_{lin}(s)(x_s) - \hat{y}(x_s)) . \tag{108}$$

It can be observed that $F_{lin}, \hat{F}, \theta_{lin}$ satisfy the following relation:

$$\forall s \in \mathbb{N}^0 : F_{lin}(s) = \hat{F}(\theta_{lin}(s)) . \tag{109}$$

A more refined relation is the one between $F(\theta_{lin})$ and $F_{lin}(\theta)$, defined for every $s = 0 \ldots S$ as follows:

$$O(F(\theta_{lin}(s)) - F_{lin}(s)) \le O\left(\frac{\varrho^2(s)}{m(n)}\right) , \tag{110}$$

where $\varrho$ is defined as:

**Definition F.1** (Typical Linear Cumulative Deviation). We define the typical linear cumulative deviation as the bound of the cumulative deviation of $F_{lin}$ from $\hat{y}$:

$$O(\varrho(s)) = \sum_{s'=0}^{s-1} O(C'(F_{lin}(s') - \hat{y})) , \tag{111}$$

and in our case:

$$O(\varrho(s)) \le O\left(\frac{1 - e^{-\frac{s}{T}}}{1 - e^{-\frac{1}{T}}}\right) \le O(1) . \tag{112}$$

This implies that $\varrho(s) = o(m(n))$, which is essential for proving (110). We will not provide this proof here, as we will not use it directly in the remainder of this paper, and we will soon prove many similar identities.

## F.2 SMALL PERTURBATION FROM THE LINEAR SOLUTION

The initial approach of the proof aimed to demonstrate that $F$ only deviates slightly from $F_{lin}$, and that its derivatives also deviate slightly from their initial values. The intention was to use induction to show that this holds at each time step. This method is effective if the goal is merely to prove that $F$ converges to $F_{lin}$ at a rate of $O\left(\frac{1}{m(n)}\right)$ for a fixed time step. However, it poses challenges when attempting to understand how the two functions deviate from each other over time. This is due to the necessity of simultaneously tracking the evolution of all derivatives and the changes in correlations over time, which is nearly impossible.

To circumvent this issue, rather than tracking all derivatives, we will calculate how $F(\theta(s))$ deviates from $F_{lin}(s)$ by utilizing a similar relationship to the one we discovered between $\theta_{lin}$ and $F_{lin}$. This will allow us to establish bounds on $F - F_{lin}$. Although the two approaches are equivalent, and the first one is more intuitively clear, the second approach simplifies the analysis by focusing on a single object, $F - F_{lin}$.

In the following lemma, we demonstrate how a small perturbation at a given step ($s = 0 \ldots S - 1$) results in a small perturbation at the subsequent step ($s + 1$). Then, we will use these results to inductively show the deviation over time between the hypothesis function and its linear approximation.

We denote:
$$\delta(s) = F(\theta(s)) - F_{lin}(s) , \quad \eta^{\frac{1}{2}}\zeta(s) = \theta(s) - \theta_{lin}(s) , \tag{113}$$
and assume that the deviation from linearity is small, hence:

$$O(\delta(s)) \le O\left(\frac{f(s)}{m(n)}\right) , \quad O(\zeta(s)) \le O(g(s))\gamma , \tag{114}$$

where

$$f\left(s\right), g\left(s\right)^2, \varrho\left(s\right)^2 = o\left(m\left(n\right)\right) . \tag{115}$$

For some parts of our lemma, it will also be relevant to separate the deviation of the parameters into two components:

$$\zeta\left(s\right) = \zeta_\gamma\left(s\right) + \zeta_m\left(s\right) , \tag{116}$$

such that:

$$O\left(\zeta_\gamma\left(s\right)\right) \le O\left(g_\gamma\left(s\right)\right)\gamma, \ O\left(\zeta_m\left(s\right)\right) \le O\left(\frac{g_m\left(s\right)}{m\left(n\right)}\right) . \tag{117}$$

**Remark F.1.** Here, we consider the case of a general rate of convergence for $\mathcal{C}'\left(F_{lin}, \hat{y}\right)$, rather than exclusively focusing on an exponential one. This is done to simplify the generalization of our results for the reader.

**Remark F.2.** In the following lemma and its proof, we use the symbol "$\simeq$" to denote higher-order terms of the expressions. This is justified by our assumption that we are working within the framework of analytic functions, where the sum of all higher-order terms still converges.

**Lemma F.1** (Deviation of the Parameters and of the Hypothesis Function Relationships). Given the conditions described above, then up to the leading order:

1.

$$\delta\left(s\right) = F\left(\theta\left(s\right)\right) - F_{lin}\left(s\right) \simeq \eta^{\frac{1}{2}}\nabla F\left(\theta_0\right)^T \zeta_m\left(s\right) + \eta^{\frac{1}{2}}\nabla F\left(\theta_0\right)^T \zeta_\gamma\left(s\right) +$$
$$\sum_{s_1,s_2=0}^{s-1} \mathfrak{C}^2 \mathcal{C}'\left(F_{lin}\left(s_1\right),\hat{y}\right) \times \mathcal{C}'\left(F_{lin}\left(s_2\right),\hat{y}\right) + \tag{118}$$
$$2\sum_{s'=0}^{s-1} \mathfrak{C}^{1,1}\zeta_\gamma\left(s\right)\mathcal{C}'\left(F_{lin}\left(s'\right),\hat{y}\right) + \eta\nabla^{\times 2}F\left(\theta_0\right)^T \zeta_\gamma\left(s\right)^{\times 2} ,$$

which means:

$$O\left(\delta\left(s\right)\right) = O\left(F\left(\theta\left(s\right)\right) - F_{lin}\left(s\right)\right) \le$$
$$O\left(\frac{g_m(s)}{m(n)}\right) + O\left(\frac{(g_\gamma(s)+\varrho(s))^2}{m(n)}\right) \le O\left(\frac{(g(s)+\varrho(s))^2}{m(n)}\right) . \tag{119}$$

2.

$$O\left(\eta^{\frac{1}{2}}\nabla F\left(\theta\left(s\right)\right)^T - \eta^{\frac{1}{2}}\nabla F\left(\theta_0\right)^T\right) \le O\left(g\left(s\right)+\varrho\left(s\right)\right)\gamma . \tag{120}$$

3.

$$\mathcal{C}'\left(F\left(\theta\left(s\right)\right),\hat{y}\right) - \mathcal{C}'\left(F_{lin}\left(s\right),\hat{y}\right) \simeq \mathcal{C}''\left(F_{lin}\left(s\right),\hat{y}\right)\delta\left(s\right) . \tag{121}$$

where $\mathcal{C}''\left(F_{\mathrm{lin}}(s),\hat{y}\right)$ denotes a positive random matrix such that, if the asymptotic behavior of $\mathcal{C}'\left(F_{\mathrm{lin}}(s),\hat{y}\right)$ is bounded, then $\mathcal{C}''\left(F_{\mathrm{lin}}(s),\hat{y}\right)$ is bounded as well (as in our setting).

4.

$$\eta^{\frac{1}{2}}\zeta\left(s+1\right) - \eta^{\frac{1}{2}}\zeta\left(s\right) = \theta\left(s+1\right) - \theta_{lin}\left(s+1\right) - \eta^{\frac{1}{2}}\zeta\left(s\right) \simeq$$
$$-\eta\nabla F\left(\theta_0\right)\mathcal{C}''\left(F_{lin}\left(s\right),\hat{y}\right)\delta\left(s\right) + O\left(g\left(s\right)+\varrho\left(s\right)\right)\mathcal{C}'\left(F_{lin}\left(s\right),\hat{y}\right)\eta^{\frac{1}{2}}\gamma , \tag{122}$$

which means:

$$O\left(\zeta\left(s+1\right)-\zeta\left(s\right)\right) \le O\left(\frac{f\left(s\right)}{m\left(n\right)}\right) + O\left(\mathcal{C}'\left(F_{lin}\left(s\right),\hat{y}\right)\right)O\left(g\left(s\right)+\varrho\left(s\right)\right)\gamma . \tag{123}$$

5.

$$O\left(\delta\left(s+1\right) - \delta\left(s\right) + \Theta_0\mathcal{C}''\left(F_{lin}\left(s\right),\hat{y}\right)\delta\left(s\right)\right) \le$$
$$O\left(\frac{(g(s)+\varrho(s))^2}{m(n)}\right)O\left(\mathcal{C}'\left(F_{lin}\left(s\right),\hat{y}\right)\right) . \tag{124}$$

**Remark F.3.** An important note for our proofs is that all of these components can be generalized to the case where $\zeta(s), \delta(s)$ are not the "original" deviations, as long as they satisfy Equation 114.

We can now use this result to prove Corollary 4.1 by induction. In fact, for the conditions of the corollary at $s = 0$, the induction hypothesis is trivially satisfied as $F(\theta)(0) = F_{lin}(0)$ and $\theta(0) = \theta_{lin}(0)$. It is straightforward to show that the contributions of the part multiplied by $O\left(\mathcal{C}'\left(F_{lin}\left(s\right),\hat{y}\right)\right)$ are irrelevant for the possible deviation, as $\mathcal{C}'\left(F_{lin}\left(s\right),\hat{y}\right) \to 0$ and $O\left(\varrho\left(s\right)\right) \leq O\left(1\right)$. Consequently, we are left with equations of motion for the asymptotic behavior of the form:

$$O\left(\zeta\left(s+1\right) - \zeta\left(s\right)\right) \leq O\left(\frac{f\left(s\right)}{m\left(n\right)}\right) \quad \delta\left(s+1\right) - \delta\left(s\right) + \Theta_0 \mathcal{C}''\left(F_{lin}\left(s\right),\hat{y}\right)\delta\left(s\right) \simeq 0 \,. \quad (125)$$

However, $\Theta_0$ and $\mathcal{C}''$ are positive definite and bounded matrices, so for a learning rate that is sufficiently small (which would be of the same order of magnitude as the learning rate needed for our system to consistently learn, and for the case where $\mathcal{C}(x) = \frac{1}{2}x^2$, exactly the same), we find that on average this term can only contribute to the shrinkage of $\delta(s)$. This means that neglecting this term for large $s$ provides an upper bound for the rate of deviation. Thus, we find that the asymptotic behavior of $\delta$ (and consequently, $\zeta$) with respect to time for large $s$ is bounded by:

$$\delta\left(s+1\right) - \delta\left(s\right) \simeq 0 \,. \quad (126)$$

**This proves our corollary.**

*Proof*.

Part - (1):

$$F\left(\theta\left(s\right)\right) = F\left(\theta_{lin}\left(s\right) + \eta^{\frac{1}{2}}\zeta\left(s\right)\right) =_1$$
$$F\left(\theta_0 - \eta\sum_{s'=0}^{s-1}\nabla F\left(\theta_0\right)\mathcal{C}'\left(F_{lin}\left(s'\right),\hat{y}\right) + \eta^{\frac{1}{2}}\zeta\left(s\right)\right) =_2$$
$$F\left(\theta_0\right) - \sum_{s'=0}^{s-1}\mathfrak{C}^1\mathcal{C}'\left(F_{lin}\left(s'\right),\hat{y}\right) + \eta^{\frac{1}{2}}\nabla F\left(\theta_0\right)^T\zeta\left(s\right) +$$
$$\sum_{s_1,s_2=0}^{s-1}\mathfrak{C}^2\mathcal{C}'\left(F_{lin}\left(s_1\right),\hat{y}\right)\times\mathcal{C}'\left(F_{lin}\left(s_2\right),\hat{y}\right) + \quad (127)$$
$$2\sum_{s'=0}^{s-1}\mathfrak{C}^{1,1}\zeta\left(s\right)\mathcal{C}'\left(F_{lin}\left(s'\right),\hat{y}\right) + \eta\nabla^{\times 2}F\left(\theta_0\right)^T\zeta\left(s\right)^{\times 2} + \ldots \simeq_3$$
$$F_{lin}\left(s\right) + \eta^{\frac{1}{2}}\nabla F\left(\theta_0\right)^T\zeta\left(s\right) + \sum_{s_1,s_2=0}^{s-1}\mathfrak{C}^2\mathcal{C}'\left(F_{lin}\left(s_1\right),\hat{y}\right)\times\mathcal{C}'\left(F_{lin}\left(s_2\right),\hat{y}\right) +$$
$$2\sum_{s'=0}^{s-1}\mathfrak{C}^{1,1}\zeta\left(s\right)\mathcal{C}'\left(F_{lin}\left(s'\right),\hat{y}\right) + \eta\nabla^{\times 2}F\left(\theta_0\right)^T\zeta\left(s\right)^{\times 2} \,,$$

where in (1) we used Equation 108 and the definition of $\theta_{lin}$, in (2) we expanded our generalization function as a Taylor series and used the definition of the derivatives correlations (3.1). In (3) we used the fact that under our assumptions our system is exponentially weakly correlated. Using this result, we obtain our desired identity.

Subtracting $F_{lin}$, we get, up to the leading order:
$$F\left(\theta\left(s\right)\right) - F_{lin}\left(s\right) \simeq$$
$$\eta^{\frac{1}{2}}\nabla F\left(\theta_0\right)^T\zeta\left(s\right) + \sum_{s_1,s_2=0}^{s-1}\mathfrak{C}^2\mathcal{C}'\left(F_{lin}\left(s_1\right),\hat{y}\right)\times\mathcal{C}'\left(F_{lin}\left(s_2\right),\hat{y}\right) +$$
$$2\sum_{s'=0}^{s-1}\mathfrak{C}^{1,1}\zeta\left(s\right)\mathcal{C}'\left(F_{lin}\left(s'\right),\hat{y}\right) + \eta\nabla^{\times 2}F\left(\theta_0\right)^T\zeta\left(s\right)^{\times 2} \simeq$$
$$\eta^{\frac{1}{2}}\nabla F\left(\theta_0\right)^T\zeta_m\left(s\right) + \eta^{\frac{1}{2}}\nabla F\left(\theta_0\right)^T\zeta_\gamma\left(s\right) +$$
$$\sum_{s_1,s_2=0}^{s-1}\mathfrak{C}^2\mathcal{C}'\left(F_{lin}\left(s_1\right),\hat{y}\right)\times\mathcal{C}'\left(F_{lin}\left(s_2\right),\hat{y}\right) + \quad (128)$$
$$2\sum_{s'=0}^{s-1}\mathfrak{C}^{1,1}\zeta_\gamma\left(s\right)\mathcal{C}'\left(F_{lin}\left(s'\right),\hat{y}\right) + \eta\nabla^{\times 2}F\left(\theta_0\right)^T\zeta_\gamma\left(s\right)^{\times 2} =$$
$$O\left(\frac{g_m(s)+g_\gamma(s)}{m(n)}\right) + O\left(\frac{\varrho(s)^2}{m(n)}\right) + 2O\left(\frac{\varrho(s)g_\gamma(s)}{m(n)}\right) + O\left(\frac{g_\gamma^2(s)}{m(n)}\right) =$$
$$O\left(\frac{g_m(s)}{m(n)}\right) + O\left(\frac{(g_\gamma(s)+\varrho(s))^2}{m(n)}\right) \leq O\left(\frac{(g(s)+\varrho(s))^2}{m(n)}\right) \,,$$

which finishes our proof.

Part 2:

Using the same ideas, we get:

$$\eta^{\frac{1}{2}}\nabla F\left(\theta_0\right)^T = \eta^{\frac{1}{2}}\nabla F\left(\theta_{lin}\left(s\right) + \eta^{\frac{1}{2}}\zeta\left(s\right)\right)^T =$$
$$\eta^{\frac{1}{2}}\nabla F\left(\theta_0 - \eta\sum_{s'=0}^{s-1}\nabla F\left(\theta_0\right)\mathcal{C}'\left(F_{lin}\left(s'\right),\hat{y}\right) + \eta^{\frac{1}{2}}\zeta\left(s\right)\right)^T =$$
$$\eta^{\frac{1}{2}}\nabla F\left(\theta_0\right)^T - \sum_{s'=0}^{s-1}\mathfrak{C}^{1,1}\mathcal{C}'\left(F_{lin}\left(s'\right),\hat{y}\right) + \eta\nabla^{\times 2}F\left(\theta_0\right)^T\zeta\left(s\right) + \ldots =$$
$$\eta^{\frac{1}{2}}\nabla F\left(\theta_0\right)^T + O\left(\varrho\left(s\right)\right)\gamma_t + O\left(g\left(s\right)\right)\gamma_t .$$

(129)

Taking the transpose on both sides completes the proof.

Part 3:

Using the definition of $\delta$ and the fact that $\mathcal{C}$ is analytical, we know that up to the highest order:

$$\mathcal{C}'\left(F\left(\theta\left(s\right)\right),\hat{y}\right) = \mathcal{C}'\left(F_{lin}\left(s\right) + \delta\left(s\right),\hat{y}\right) \simeq \mathcal{C}'\left(F_{lin}\left(s\right),\hat{y}\right) + \mathcal{C}''\left(F_{lin}\left(s\right),\hat{y}\right)\delta\left(s\right) . \quad (130)$$

Since $\mathcal{C}$ is convex (3), its second derivative is always a positive matrix, and if the first derivative is bounded, then so is the second.

Part 4:

Using the equation of motion for $\theta$ (108), and parts 2 and 3 of this lemma, we get, up to leading order:

$$\theta\left(s+1\right) = \theta\left(s\right) - \eta\nabla F\left(\theta\left(s\right)\right)\mathcal{C}'\left(F\left(\theta\left(s\right)\right),\hat{y}\right) \simeq$$
$$\theta\left(s\right) - \eta\begin{pmatrix} \nabla F\left(\theta_0\right) + \\ O\left(g\left(s\right) + \varrho\left(s\right)\right)\eta^{\frac{1}{2}}\gamma \end{pmatrix}\begin{pmatrix} \mathcal{C}'\left(F_{lin}\left(s\right),\hat{y}\right) + \\ \mathcal{C}''\left(F_{lin}\left(s\right),\hat{y}\right)\delta\left(s\right) \end{pmatrix} \simeq$$
$$\theta\left(s\right) - \eta\nabla F\left(\theta_0\right)\mathcal{C}'\left(F_{lin}\left(s\right),\hat{y}\right) - \eta\nabla F\left(\theta_0\right)\mathcal{C}''\left(F_{lin}\left(s\right),\hat{y}\right)\delta\left(s\right) +$$
$$O\left(g\left(s\right) + \varrho\left(s\right)\right)\mathcal{C}'\left(F_{lin}\left(s\right),\hat{y}\right)\eta^{\frac{1}{2}}\gamma ,$$

(131)

and since:

$$\theta\left(s\right) - \eta\nabla F\left(\theta\left(s\right)\right)\mathcal{C}'\left(F\left(\theta\left(s\right)\right),\hat{y}\right) =$$
$$\theta_{lin}\left(s\right) + \eta^{\frac{1}{2}}\zeta\left(s\right) - \eta\nabla F\left(\theta\left(s\right)\right)\mathcal{C}'\left(F\left(\theta\left(s\right)\right),\hat{y}\right) = \theta_{lin}\left(s+1\right) + \eta^{\frac{1}{2}}\zeta\left(s\right) ,$$

(132)

we obtain the desired result.

Part 5:

Using the equation of motion for $\theta$, one can see that:

$$F\left(\theta\left(s+1\right)\right) = F\begin{pmatrix} \theta_{lin}\left(s+1\right) - \eta\nabla F\left(\theta_0\right)\mathcal{C}''\left(F_{lin}\left(s\right),\hat{y}\right)\delta\left(s\right) + \\ \eta^{\frac{1}{2}}\zeta\left(s\right) + O\left(g\left(s\right) + \varrho\left(s\right)\right)\mathcal{C}'\left(F_{lin}\left(s\right),\hat{y}\right)\eta^{\frac{1}{2}}\gamma \end{pmatrix}$$
$$\simeq_1$$
$$F_{lin}\left(s+1\right) - \eta\nabla F\left(\theta_0\right)^T\nabla F\left(\theta_0\right)\mathcal{C}''\left(F_{lin}\left(s\right),\hat{y}\right)\delta\left(s\right) +$$
$$\eta^{\frac{1}{2}}\nabla F\left(\theta_0\right)^T\zeta\left(s\right) + O\left(g\left(s\right) + \varrho\left(s\right)\right)\mathcal{C}'\left(F_{lin}\left(s\right),\hat{y}\right)\eta^{\frac{1}{2}}\nabla F\left(\theta_0\right)^T\gamma +$$
$$\sum_{s_1,s_2=0}^{s-1}\mathfrak{C}^2\mathcal{C}'\left(F_{lin}\left(s_1\right),\hat{y}\right) \times \mathcal{C}'\left(F_{lin}\left(s_2\right),\hat{y}\right) +$$
$$2\sum_{s'=0}^{s-1}\mathfrak{C}^{1,1}\zeta\left(s\right)\mathcal{C}'\left(F_{lin}\left(s'\right),\hat{y}\right) +$$
$$2O\left(g\left(s\right) + \varrho\left(s\right)\right)\mathcal{C}'\left(F_{lin}\left(s\right),\hat{y}\right)\sum_{s'=0}^{s-1}\mathfrak{C}^{1,1}\gamma\mathcal{C}'\left(F_{lin}\left(s'\right),\hat{y}\right) +$$
$$\eta\nabla^{\times 2}F\left(\theta_0\right)^T\zeta\left(s\right)^{\times 2} + O\left(g\left(s\right) + \varrho\left(s\right)\right)^2\mathcal{C}'\left(F_{lin}\left(s\right),\hat{y}\right)^2\eta\nabla^{\times 2}F\left(\theta_0\right)^T\gamma^{\times 2} +$$
$$2O\left(g\left(s\right) + \varrho\left(s\right)\right)\mathcal{C}'\left(F_{lin}\left(s\right),\hat{y}\right)\eta\nabla^{\times 2}F\left(\theta_0\right)^T\left(\gamma \times \zeta\left(s\right)\right)$$
$$\simeq_2$$
$$F_{lin}\left(s+1\right) - \Theta_0\mathcal{C}''\left(F_{lin}\left(s\right),\hat{y}\right)\delta\left(s\right) + \delta\left(s\right) +$$
$$2O\left(\frac{g(s)+\varrho(s)}{m(n)}\right)O\left(\mathcal{C}'\left(F_{lin}\left(s\right),\hat{y}\right)\right) + O\left(\frac{(g(s)+\varrho(s))^2}{m(n)}\right)O\left(\mathcal{C}'\left(F_{lin}\left(s\right),\hat{y}\right)\right)^2 +$$
$$2O\left(\frac{g^2(s)+\varrho(s)g(s)}{m(n)}\right)O\left(\mathcal{C}'\left(F_{lin}\left(s\right),\hat{y}\right)\right) .$$

(133)

where in (1) we use part 1 of the lemma, noting that $O(\delta) \leq O\left(\frac{1}{m(n)}\right)$, so it can be considered as $\zeta_m$. In (2) we use the definition of $F_{lin}$, $\Theta_0$, and part 1 once again, where we gathered all of the components that have only $\zeta(s)$ to get $\delta(s)$. We then used the asymptotic behavior of all components and took the worst-case scenario to obtain Equation 124. $\square$

# G  WIDE NEURAL NETWORKS ARE WEAKLY CORRELATED PGDML SYSTEMS

## G.1  GENERAL IDEA

We start with fully connected neural networks. Although the proof is technically intricate, its underlying concept is straightforward: for the first layer, we observe that all higher correlations exhibit the appropriate asymptotic behavior. We then proceed by induction to show that all layers exhibit the same asymptotic behavior. Consider the second correlation, for instance, which we analyze as follows:

For any layer $l = 1, \ldots, L$, defining $\nabla_{-l}$ as the derivatives with respect to parameters from layers 1 to $l-1$ (G.2), we employ the equation for fully connected neural networks (139):

$$
\begin{aligned}
l = 0, \ldots, L : F^{(l)} = \theta^{(l,l-1)}\phi\left(F^{(l-1)}\right) + \theta^{(l)} \,, \\
\forall x \in X : F(\theta)\left(x\right) = F^{(L)}\left(x\right), \quad F^{(0)}\left(x\right) = a \,,
\end{aligned}
\tag{134}
$$

to demonstrate that:

$$
\begin{aligned}
\nabla_{(-l)}^{\times 2} F^{(l)} = \nabla_{(-l)}^{\times 2}\left(\theta^{(l,l-1)}\phi\left(F^{(l-1)}\right) + \theta^{(l)}\right) = \\
\nabla_{(-l)} \times \nabla_{(-l)}\left(\theta^{(l,l-1)}\phi\left(F^{(l-1)}\right) + \theta^{(l)}\right) = \nabla_{(-l)} \times \left(\theta^{(l,l-1)}\nabla_{(-l)}\phi\left(F^{(l-1)}\right)\right) = \\
\nabla_{(-l)} \times \left(\theta^{(l,l-1)}\phi'\left(F^{(l-1)}\right)\nabla_{(-l)}F^{(l-1)}\right) = \\
\theta^{(l,l-1)}\phi''\left(F^{(l-1)}\right)\nabla_{(-l)}F^{(l-1)} \times \nabla_{(-l)}F^{(l-1)} + \theta^{(l,l-1)}\phi'\left(F^{(l-1)}\right)\nabla_{(-l)}^{\times 2}F^{(l-1)}
\end{aligned}
\tag{135}
$$

Consequently, the contribution to the $l$-th correlation (10) from this part is proportional to:

$$
\theta^{(l,l-1)}\phi''\left(F^{(l-1)}\right)\mathfrak{C}_{(l-1)}^1 \times \mathfrak{C}_{(l-1)}^1 + \theta^{(l,l-1)}\phi'\left(F^{(l-1)}\right)\mathfrak{C}_{(l-1)}^2 \,.
\tag{136}
$$

Here we have two terms. We can show the right-hand term is small by induction. Showing that the left-hand term is also small is more involved: it requires establishing that, for all hidden layers, the relevant contribution from the first correlation originates from its diagonal terms, i.e., $\left(\mathfrak{C}_{(-l)}^1\right)_{ii}$.

We can now show that, in this term, the left index is identical for both correlations. It follows that, for most indices, the relevant terms are offset by the irrelevant ones, keeping the overall expression small.

For the case where one of the derivatives does not belong to layers 1 to $l-1$, we explicitly show the corresponding term is negligible, since for most indices it simply resets:

$$
\nabla_{i^l i^{l-1}} F_i^{(l)} \propto \delta_{i^l i}
\tag{137}
$$

In the general case of the $D$-th correlation, while tracing the combinatorial terms arising from different derivative combinations is somewhat intricate, the underlying principle remains the same.

The generalization of this approach to other architectures is discussed in Section G.5.

## G.2  ASYMPTOTIC BEHAVIOR OF WIDE FCN AT INITIALISATION

**Remark G.1.** Throughout this paper we have considered $\|M\|$ or $O(M)$ as our way to evaluate the size of our random tensors. But here we mainly consider the normalised terms instead:

$$
\frac{1}{\sqrt{N}}\|M\| \quad \text{and} \quad \frac{1}{\sqrt{N}}O\left(M\right) \,.
\tag{138}
$$

This is because, in practice, what we are interested in is the average asymptotic behavior of a tensor, and not the accumulative one.

Fully connected neural networks of depth $2 \leq L \in \mathbb{N}$, characterized by $L$ parameter vectors (the biases $\theta^{(1)}, \ldots, \theta^{(L)}$) and $L$ parameter matrices (the weights $\theta^{(L,L-1)}, \ldots, \theta^{(1,0)}$), such as:

$$
\begin{aligned}
l = 0, \ldots, L : F^{(l)} &= \theta^{(l,l-1)} \phi \left( F^{(l-1)} \right) + \theta^{(l)} , \\
\forall x \in X : F(\theta)(x) &= F^{(L)}(x), \quad F^{(0)}(x) = a .
\end{aligned}
\tag{139}
$$

In this representation, $F^{(0)}, F^{(1)}, \ldots, F^{(L-1)}$, and $F^{(L)}$ constitute the input, inner, and output layers, respectively. The activation function $\phi$ is analytical, and all of its derivatives are bounded as described in (23).

**Remark G.2.** Generally, when working with FCNNs we do not apply the activation function to the zeroth layer (the input). However, to make the induction slightly easier, we will simplify the expression by assuming that $\phi$ operates on all layers. This makes no material difference.

We focus on "wide" neural networks where the depth $L$ is fixed. As long as $L = O(\log(n))$, we can expect NTK-like behavior for large $n$, but for simplicity we focus on the scenario where $L$ remains constant in $n$. We introduce a limiting parameter $n \in \mathbb{N}$ such that the width of all hidden layers satisfies $n \leq n_1, \ldots, n_{L-1}$. To simplify our work, we replace this assumption by postulating that all hidden layers exhibit the same asymptotic behavior in $n$, i.e., $n_1, \ldots, n_{L-1} \sim n$. This modification does not affect our theorems and lemmas, as it merely establishes a lower bound implied by the original assumption. Since the sizes of the zeroth and last layers are constant (the input and output dimensions remain fixed in $n$), we obtain:

$$
n_1, \ldots, n_{L-1} \sim n \quad \text{and} \quad n_0, n_L \sim 1 .
\tag{140}
$$

Back in the 1960s, it was demonstrated that with Gaussian initialization, we can keep our layers normalised by selecting initial parameters as follows:

$$
\forall l = 1, \ldots, L : \theta_0^{(l,l-1)} \sim \mathcal{N}\left( 0, \frac{1}{n_l} \right), \, \theta_0^{(l)} \sim \mathcal{N}(0, 1) .
\tag{141}
$$

Despite the specificity of this initialization algorithm, it contradicts the broader spirit of this paper. It is not only overly restrictive, but it also complicates our work by colliding with our framework for tensor asymptotic behavior. Rather than focusing on a particular initialization scheme such as the normal distribution, we identify and use the relevant properties of the distribution.

**Definition G.1** (Appropriate Initialization Scheme for Wide Neural Networks). Given a wide neural network as defined above, we characterize the distribution for the initial condition $\theta$ as appropriate if and only if, for every probability arbitrarily close to 1, the following properties hold:

1. Different elements of $\theta$ are independent. Moreover, for each layer $l = 1, \ldots, L$, the elements of $\theta^{(l,l-1)}$ and of $\theta^{(l)}$ share the same distribution within their respective tensors.

2. $\theta$ is symmetric around 0 (implying that all odd moments vanish):
$$
\forall D \in \mathbb{N} \setminus 2\mathbb{N} : \left\langle \theta^{\cdot D} \right\rangle = 0 .
\tag{142}
$$

3. For every layer $l = 1, \ldots, L$, all moments of $\theta$ are uniformly normalized:
$$
\forall D \in \mathbb{N} : \quad
\begin{aligned}
O(1)^D &\leq \frac{1}{\sqrt{n_l}} O\left( \left(\theta^{(l)}\right)^{\cdot D} \right) \leq D! O(1)^D , \\
O\left(\frac{1}{\sqrt{n_{l-1}}}\right)^D &\leq \frac{1}{\sqrt{N_l}} O\left( \left(\theta^{(l,l-1)}\right)^{\cdot D} \right) \leq D! O\left(\frac{1}{\sqrt{n_{l-1}}}\right)^D ,
\end{aligned}
\quad \text{Uniformly}
\tag{143}
$$
where $N_l = n_l n_{l-1}$ is the total number of parameters in the $l$-th layer.

where the elemental tensor power is defined by:
$$
\forall D \in \mathbb{N} : \left( M^{\cdot D} \right)_{\vec{i}} = M_{\vec{i}}^D .
\tag{144}
$$

The first two conditions ensure that our system is unbiased, while the third condition guarantees that it is not dominated by a disproportionate probabilistic "tail."

We delegate to the reader the verification that Gaussian initialization qualifies as an appropriate initialization.

**Remark G.3.** Conditions 1 and 2 can be generalized to hold in the limit of large $n$, provided this convergence occurs rapidly enough. Nevertheless, any complexities arising from this generalization are technical and do not affect our analysis.

**For the remainder of this section, we will omit the biases from our discussion, as they do not add any substantial insights or implications for the points under consideration and do not change any of our results.**

**Lemma G.1** (Normalization of Layers in Proper Wide Neural Networks)**.** Given a wide neural network, if the initial condition is appropriately set, then all the moments across every layer $l = 1, \ldots, L$ are well normalized:

$$\frac{1}{\sqrt{n_l}} O\left(F^{(l)}\right) = O(1) . \tag{145}$$

The final parameter that we need to normalize in our system is the dynamic one-the learning rate, denoted by $\eta$. In an attempt to generalize Gaussian initialization, we adopt the standard normalization for $\eta$:

$$\eta \sim \frac{1}{n} . \tag{146}$$

This condition, coupled with the demand for an appropriate initialization strategy, is sufficient to demonstrate that wide neural networks are exponentially weakly correlated PGDML systems.

**In the remainder of this section, we will proceed under the assumption that our parameters are initialized appropriately and that $\eta \sim \frac{1}{n}$.**

We can now use this result to find the asymptotic behavior of the layer derivatives:

**Lemma G.2** (Asymptotic Behavior of Layer Derivatives)**.** Given our established conditions and initialisation, all derivatives are uniformly bounded for each natural number $D$ and each layer $l = 1, \ldots, L$. Specifically, we have:

$$\frac{\eta^{\frac{D}{2}}}{\sqrt{N_D}} O\left(\nabla^{\times D} F^{(l)}\right) \leq O(1) \quad \text{Uniformly} \tag{147}$$

Here, $N_D = n_l n_{l-1}^D n^D$ represents the asymptotic behavior of the number of elements in the derivatives.

### *Proof of Lemma G.1.*

We approach the proof by induction. Throughout the proof we use Lemma C.7 to establish equivalence between the asymptotic behavior of the system and its tensorial average (79). The base case, the zeroth layer, satisfies the lemma trivially. Assume inductively that layer $l - 1$ satisfies the lemma. We then prove the lemma for the $l$-th layer for all $l = 1, \ldots, L$:

$$\left[F^{(l)}\right]^2 =_1 \frac{1}{n_l} \sum_i \left\langle \left(\sum_j \theta_{ij}^{(l,l-1)} F_j^{(l-1)}\right)\left(\sum_k \theta_{ik}^{(l,l-1)} F_k^{(l-1)}\right)\right\rangle =$$
$$\frac{1}{n_l} \sum_i \left\langle \left(\sum_{j,k} \theta_{ij}^{(l,l-1)} \theta_{ik}^{(l,l-1)} F_j^{(l-1)} F_k^{(l-1)}\right)\right\rangle =_2$$
$$\frac{1}{n_l} \sum_i \sum_{j,k} \left\langle \theta_{ij}^{(l,l-1)} \theta_{ik}^{(l,l-1)}\right\rangle \left\langle F_j^{(l-1)} F_k^{(l-1)}\right\rangle =$$
$$\frac{1}{n_l} \sum_i \sum_{j \neq k} \left\langle \theta_{ij}^{(l,l-1)} \theta_{ik}^{(l,l-1)}\right\rangle \left\langle F_j^{(l-1)} F_k^{(l-1)}\right\rangle + \frac{1}{n_l} \sum_i \sum_j \left\langle \left(\theta_{ij}^{(l,l-1)}\right)^2\right\rangle \left\langle \left(F_j^{(l-1)}\right)^2\right\rangle =_3$$
$$\sum_i \sum_j \frac{1}{n_l} \left\langle \left(\theta_{ij}^{(l,l-1)}\right)^2\right\rangle \left\langle \left(F_j^{(l-1)}\right)^2\right\rangle =_4 \sum_{i,j} \frac{1}{n_l} \left\langle \left(\theta_{ij}^{(l,l-1)}\right)^2\right\rangle \sum_k \frac{1}{n_{l-1}} \left\langle \left(F_k^{(l-1)}\right)^2\right\rangle =_5$$
$$n_{l-1} \left[\theta^{(l,l-1)}\right]^2 \left[F^{(l-1)}\right]^2 =_6 O(1) O(1) = O(1) . \tag{148}$$

Throughout these equalities, we rely on the premise of a proper initialization. Specifically:

- In "1" and "5", we employ the structure of neural networks and the definition of the moment norm.

- In "2" and "4", we note that $F^{(l-1)}$ depends only on the inner parameters of $l$, which are independent of $\theta^{(l,l-1)}$. This is enabled by the proper initialization ensuring $\theta^{(l,l-1)}$ is uniformly distributed.

- In "3", we invoke the fact that different elements of $\theta^{(l,l-1)}$ are independent and symmetric. Hence, for every $i$ and $j \neq k$:

$$\left\langle \theta_{ij}^{(l,l-1)} \theta_{ik}^{(l,l-1)} \right\rangle = \left\langle \theta_{ij}^{(l,l-1)} \right\rangle \left\langle \theta_{ik}^{(l,l-1)} \right\rangle = 0 . \tag{149}$$

- In "6", we apply the induction hypothesis and observe that for a proper initialization (G.1-3):

$$\forall l = 1, \ldots, L : \left[ \theta^{(l,l-1)} \right] = O \left( \frac{1}{\sqrt{n_{l-1}}} \right) . \tag{150}$$

By mathematical induction, the lemma holds for all $l = 1, \ldots, L$.

Using Lemma C.7 again, we obtain $O\left(F^l\right) \leq O\left(1\right)$. Since the proof should still hold even after excluding an event of sufficiently small probability, we conclude that:

$$O\left(F^l\right) = O\left(1\right) . \tag{151}$$

exactly.

$\square$

***Proof of Lemma G.2.*** Given $\omega$, drawn from another proper initialisation, we observe that $\theta + \omega$ is also properly initialised (or at least sub-properly initialised). Hence, assuming we initialise $F^{(l)}$ accordingly, we find:

$$\frac{1}{\sqrt{n_l}} O\left(F^{(l)}\right) \leq O\left(1\right) . \tag{152}$$

Since $F^{(l)}$ is analytical, we can apply its Taylor expansion around $\theta$ to get:

$$\frac{1}{\sqrt{n_l}} O\left( \sum_{D=0}^{\infty} \nabla^{\times D} F^{(l)}\left(\theta\right) \omega^{\times D} \right) \leq O\left(1\right) . \tag{153}$$

By continuously rescaling $\omega$ without violating the proper property, we see that all components of the expression must be uniformly bounded:

$$\forall D \in \mathbb{N} : \frac{1}{\sqrt{n_l}} O\left( \nabla^{\times D} F^{(l)}\left(\theta\right) \omega^{\times D} \right) \leq O\left(1\right) \quad \text{Uniformly.} \tag{154}$$

This is because all terms scale differently with $\omega$, meaning the only way to ensure the expression remains bounded under any finite rescaling of $\omega$ is to bound each term separately and uniformly.

Considering the symmetry of the derivative in its components, and invoking Lemma C.3, we can identify a vector of size 1 that maximises it, yielding a vector with size equal to its norm. By setting $\omega$ to be this vector and rescaling it to be proper, we obtain, using Lemma C.7, that:

$$\forall D \in \mathbb{N} : \frac{1}{\sqrt{n_l}} O\left( \nabla^{\times D} F^{(l)}\left(\theta\right) \right) = \frac{1}{\sqrt{n_l}} \frac{1}{\sqrt{n_{l-1}^D}} O\left( \nabla^{\times D} F^{(l)}\left(\theta\right) \omega^{\times D} \right) \leq O\left(1\right) \quad \text{Uniformly.} \tag{155}$$

Given that:

$$\frac{1}{\sqrt{n_l}} \frac{1}{\sqrt{n_{l-1}^D}} = \frac{1}{\sqrt{n_l n_{l-1}^D}} \sim \frac{\eta^{\frac{D}{2}}}{\sqrt{n_l n_{l-1}^D n^D}} = \frac{\eta^{\frac{D}{2}}}{\sqrt{N_D}} , \tag{156}$$

we arrive at the desired result.

$\square$

### G.3 Representation of the Network's Layers as a Composition of Previous Layer Components

In this part, we use the semilinear structure of wide neural networks to establish a linear relation between the correlations of the $l$-th layer and those of the $l-1$-st layer. We then use this relation in the next part to show, by induction, that the correlations are weak. To that end, we define the following useful notation:

**Definition G.2** (Inner and Outer Derivatives). Given a layer $l = 1, \ldots, L$, we denote the $l$-th layer's outer parameters, which include its weights (and biases), as follows:

$$\theta_{i^l, i^{l-1}}^{(l, l-1)} . \tag{157}$$

Meanwhile, the inner parameters are defined as any of the weights (and biases) from layers $1, \ldots, l-1$, and are denoted by:

$$\theta \in \theta^{(-l)} . \tag{158}$$

Following the same notation, we denote the gradient with respect to the outer parameters by $\nabla_{(l)}$, and the gradient with respect to the inner parameters by $\nabla_{(-l)}$. The same applies to the correlations, denoted by $\mathfrak{C}_{(l)}$ and $\mathfrak{C}_{(-l)}$.

**Remark G.4.** It is important to note that, as $F^{(l-1)}$ depends only on the inner parameters of the $l$-th layer, the following relationship holds:

$$\nabla_{(-l)} F^{(l-1)} = \nabla F^{(l-1)} . \tag{159}$$

This notation can be employed to express the derivative of the $l$-th layer as a combination of derivatives from the $(l-1)$-st layer.

**Lemma G.3** (Representation of the $l$-th Layer Derivative as a Combination of the Previous Layer Derivatives). Given a fully connected wide neural network as specified above, for each layer $l = 1, \ldots, L$, the $D \in \mathbb{N}$-th derivative can be presented as follows:

1. When all derivatives are inner, we have:

$$\left( \nabla^{(-l)} \right)^{\times D} F^{(l)} = \theta^{(l, l-1)} \tilde{\nabla}^{\times D} F^{(l-1)} . \tag{160}$$

2. When one derivative is outer and the rest are inner, we have:

$$\nabla_{i_l i_{l-1}}^{(l)} \times \left( \nabla^{(-l)} \right)^{\times (D-1)} F_i^{(l)} = \delta_{i i_l} \tilde{\nabla}^{\times (D-1)} F_{i_{l-1}}^{(l-1)} . \tag{161}$$

3. When $D \geq 2$ and $2 \leq d \in \mathbb{N} \leq D$ derivatives are outer, we have:

$$\left( \nabla^{(l)} \right)^{\times d} \times \left( \nabla^{(-l)} \right)^{\times (D-d)} F^{(l)} = 0 . \tag{162}$$

Here, $\tilde{\nabla}^{\times D} F^{(l-1)}$ denotes the compound derivative, defined as follows for $D \in \mathbb{N}$:

$$\tilde{\nabla}^{\times D} F^{(l)} = \sum_{d=1}^{D} \sum_{\substack{d_1 + \ldots + d_d = D \\ d_1 \ldots d_d \in \mathbb{N}}} \phi^{[d]} \left( F^{(l)} \right) \left( \nabla^{\times d_1} F^{(l)} \times \cdots \times \nabla^{\times d_d} F^{(l)} \right) + \text{comb} \tag{163}$$

and for $D = 0$:

$$\tilde{\nabla}_{ij}^{\times 0} F_k^{(l)} = \delta_{ik} \phi \left( F_j \right) . \tag{164}$$

The "comb" term refers to all possible combinations of the derivatives' indices. For instance, if we consider one term of the third derivative:

$$\theta^{(l, l-1)} \left( \phi^{[2]} \left( F^{(l-1)} \right) \left( \nabla F^{(l-1)} \times \nabla^{\times 2} F^{(l-1)} \right) \right) \tag{165}$$

then, for any three distinct derivative indices $\alpha_1, \alpha_2, \alpha_3$, there are three unique ways to assign the indices (disregarding irrelevant parts):

$$\nabla_{\alpha_1} F^{(l-1)} \times \nabla_{\alpha_2 \alpha_3}^{\times 2} F^{(l-1)}, \ \nabla_{\alpha_2} F^{(l-1)} \times \nabla_{\alpha_1 \alpha_3}^{\times 2} F^{(l-1)}, \ \nabla_{\alpha_3} F^{(l-1)} \times \nabla_{\alpha_1 \alpha_2}^{\times 2} F^{(l-1)} . \tag{166}$$

While the first combination naturally arises from our expression, the "comb" term accounts for the other two.

It should be mentioned that only unique terms are counted, even if they arise from different orderings of the derivatives. Therefore, for another component of the third derivative, $\theta^{(l,l-1)} \left( \phi^{[3]} \left( F^{(l-1)} \right) \nabla F^{(l-1)} \times \nabla F^{(l-1)} \times \nabla F^{(l-1)} \right)$, and distinct $\alpha_1, \alpha_2, \alpha_3$:

$$\nabla_{\alpha_1} F^{(l-1)} \nabla_{\alpha_2} F^{(l-1)} \nabla_{\alpha_3} F^{(l-1)}, \ \nabla_{\alpha_1} F^{(l-1)} \nabla_{\alpha_3} F^{(l-1)} \nabla_{\alpha_2} F^{(l-1)} \dots \tag{167}$$

are identical, and thus should only be counted once.

We can use this result to construct the $l$-th layer correlations using the correlations from the $(l-1)$-st layer:

**Lemma G.4** (Representation of the $l$-th Layer Correlations as a Combination of the Previous Layer Correlations). Under the same conditions as in Lemma G.3, we have:

$$\mathfrak{C}_{(l)}^{D,d} = \theta^{(l,l-1)} \times \left( \tilde{\theta}^{(l,l-1)} \right)^{\times d} \tilde{\mathfrak{C}}_{(l-1)}^{D,d} +$$
$$\eta^{\frac{1}{2}} I \times \eta^{\frac{1}{2}} \phi \left( F^{(l-1)} \right) \times \left( \tilde{\theta}^{(l,l-1)} \right)^{\times (d-1)} \tilde{\mathfrak{C}}_{(l-1)}^{D,d-1} + \text{comb} + \tag{168}$$
$$\left( \tilde{\theta}^{(l,l-1)} \right)^{\times d} \hat{\mathfrak{C}}_{(l-1)}^{D-1,d} + \text{comb} .$$

or, when showing the indices explicitly (Einstein summation convention):

$$\left( \mathfrak{C}_{(l)}^{D,d} \right)_{i_0 i_1 \dots i_d} = \theta_{i_0 j_0}^{(l,l-1)} \tilde{\theta}_{i_1 j_1}^{(l,l-1)} \cdots \tilde{\theta}_{i_d j_d}^{(l,l-1)} \left( \tilde{\mathfrak{C}}_{(l-1)}^{D,d} \right)_{j_0, j_1 \dots j_d} +$$
$$\eta^{\frac{1}{2}} \delta_{i_0 i_1} \eta^{\frac{1}{2}} \phi \left( F_{j_0}^{(l-1)} \right) \tilde{\theta}_{i_2 j_2}^{(l,l-1)} \cdots \tilde{\theta}_{i_d j_d}^{(l,l-1)} \left( \tilde{\mathfrak{C}}_{(l-1)}^{D,d-1} \right)_{j_0, j_2 \dots j_d} + \text{comb} + \tag{169}$$
$$\tilde{\theta}_{i_1 j_1}^{(l,l-1)} \cdots \tilde{\theta}_{i_d j_d}^{(l,l-1)} \left( \hat{\mathfrak{C}}_{(l-1)}^{D-1,d} \right)_{i_0, j_1 \dots j_d} ,$$

where the "comb" term includes all index pairings with the zero index, i.e., $(i_0, i_2), \dots, (i_0, i_d)$, and $\tilde{\theta}$ is defined by:

$$\tilde{\theta}_{ij}^{(l,l-1)} = \theta_{ij}^{(l,l-1)} \phi' \left( F_j^{(l-1)} \right) . \tag{170}$$

The first compound correlation is defined as follows for $D \in \mathbb{N}_0$ and $d \in \mathbb{N}$:

$$\tilde{\mathfrak{C}}_{(l)}^{D,d} = \sum_{d'=1}^{D+d} \left\{ C_{\vec{d},\vec{D}} \phi^{[d']} \left( F^{(l)} \right) \mathfrak{C}_{(l)}^{D_1, d_1} \times \cdots \times \mathfrak{C}_{(l)}^{D_{d'}, d_{d'}} \ \middle| \ \begin{array}{c} d_1 + \dots + d_{d'} = d \\ D_1 + \dots + D_{d'} = D \end{array} \right\} + \text{Comb} \tag{171}$$

where:

$$C_{\vec{d},\vec{D}} = \frac{(D_1! \cdots D_{d'}!) (d_1! \cdots d_{d'}!)}{D! d!} . \tag{172}$$

Also, for $D \in \mathbb{N}_0$ and $d = 0$:

$$\tilde{\mathfrak{C}}_{(l)}^{D,0} = \eta^{\frac{D}{2}} \tilde{\nabla}_t^{\times D} F^{(l)} . \tag{173}$$

The second compound correlation is defined as follows for $D \in \mathbb{N}$ and $d \in \mathbb{N}$:

$$\left( \hat{\mathfrak{C}}_{(l)}^{D-1,d} \right)_{i_0, j_1 \dots j_d}^{\alpha_{d+1} \dots \alpha_{d+D}} = \eta^{\frac{1}{2}} \delta_{(i_0 j_0)}^{\alpha_{d+1}} \left( \tilde{\mathfrak{C}}_{(l)}^{D-1,d} \right)_{j_0, j_1 \dots j_d}^{\alpha_{d+2} \dots \alpha_{d+D}} + \text{comb} , \tag{174}$$

where the "comb" term is defined as before. For $D = 0$, this compound correlation vanishes.

**Remark G.5.** For the following lemma and the subsequent section, we assume that $D \ll n$. This assumption is permissible even though, in taking limits, the limit in $D$ should technically be taken prior to that in $n$. This is because higher-order derivatives typically exert a decreasing influence on system behavior, leading us to treat them as negligible beyond a certain order.

This assumption is not strictly necessary: we could instead address the combinatorial factors directly. Nevertheless, we adopt it to avoid introducing unnecessary complications into our analysis.

**Lemma G.5** (Counting Combinations of Derivatives and Correlations).

1. For the conditions of Lemma G.3, for every $d_1, \ldots, d_d$, the number of combinations of the derivative indices is:

$$\frac{1}{d!} \frac{D!}{d_1! \cdots d_d!} \, , \tag{175}$$

and the total number of combinations over all $d = 1, \ldots, D$ is the $D$-th Bell number (which is very close to $D!$).

2. For the conditions of Lemma G.4, for every $d_1, \ldots, d_{d'}$ and $D_1, \ldots, D_{d'}$, the number of combinations of the compound correlations is:

$$\frac{1}{d'!} \frac{d!}{d_1! \cdots d_{d'}!} \frac{D!}{D_1! \cdots D_{d'}!} \, . \tag{176}$$

We assume for this lemma that the indices are different, as $D \ll n$.

*Proof - Lemmas G.3 and G.4*.

We prove the lemma by induction for a general layer $l = 1, \ldots, L - 1$, starting with $l = 1$.

The induction base is simple: this is a direct consequence of differentiating the network equation (139). This calculation hinges on the fact that, by definition, the inner derivatives are independent of the outer parameters:

$$\nabla_{(l)} F^{(l)} = \nabla_{(l)} \theta^{(l,l-1)} \phi \left( F^{(l-1)} \right) = \theta^{(l,l-1)} \nabla_{(l)} \phi \left( F^{(l-1)} \right) = \\ \theta^{(l,l-1)} \left( \phi^{[1]} \left( F^{(l-1)} \right) \nabla_{(l)} F^{(l-1)} \right) \, , \tag{177}$$

which yields the base case.

Assume by induction that the lemma holds for some $D - 1 \in \mathbb{N}$: the inner $D$-th derivative satisfies:

$$\nabla_{(-l)}^{\times D} F^{(l)} = \nabla_{(-l)} \times \nabla_{(-l)}^{\times (D-1)} F^{(l)} = \\ \nabla_{(-l)} \times \theta^{(l,l-1)} \sum_{d=1}^{D-1} \sum_{d_1 \ldots d_d \in \mathbb{N}}^{d_1 + \ldots + d_d = D-1} \phi^{[d]} \left( F^{(l-1)} \right) \left( \nabla^{\times d_1} F^{(l-1)} \times \cdots \times \nabla^{\times d_d} F^{(l-1)} \right) \\ + \text{comb} \\ = \\ \theta^{(l,l-1)} \sum_{d=1}^{D-1} \sum_{d_1 \ldots d_d \in \mathbb{N}}^{d_1 + \ldots + d_d = D-1} \nabla \times \phi^{[d]} \left( F^{(l-1)} \right) \left( \nabla^{\times d_1} F^{(l-1)} \times \cdots \times \nabla^{\times d_d} F^{(l-1)} \right) \\ + \\ \theta^{(l,l-1)} \sum_{d=1}^{D-1} \sum_{d_1 \ldots d_d \in \mathbb{N}}^{d_1 + \ldots + d_d = D-1} \phi^{[d]} \left( F^{(l-1)} \right) \left( \nabla \times \nabla^{\times d_1} F^{(l-1)} \times \cdots \times \nabla^{\times d_d} F^{(l-1)} \right) \\ + \\ \text{comb} \tag{178}$$

We thus obtain a sum of two different contributions; we analyze each separately.

Starting with the first:

$$\sum_{d=1}^{D-1} \sum_{d_1 \ldots d_d \in \mathbb{N}}^{d_1 + \ldots + d_d = D-1} \nabla \times \phi^{[d]} \left( F^{(l-1)} \right) \left( \nabla^{\times d_1} F^{(l-1)} \times \cdots \times \nabla^{\times d_d} F^{(l-1)} \right) \\ = \\ \sum_{d=1}^{D-1} \sum_{d_1 \ldots d_d \in \mathbb{N}}^{d_1 + \ldots + d_d = D-1} \phi^{[d+1]} \left( F^{(l-1)} \right) \left( \nabla F^{(l-1)} \times \nabla^{\times d_1} F^{(l-1)} \times \cdots \times \nabla^{\times d_d} F^{(l-1)} \right) \\ = \\ \sum_{d=1}^{D-1} \sum_{d_1 = 1, d_2 \ldots d_{d+1} \in \mathbb{N}}^{d_1 + d_2 + \ldots + d_{d+1} = D} \phi^{[d+1]} \left( F^{(l-1)} \right) \left( \nabla^{\times d_1} F^{(l-1)} \times \nabla^{\times d_2} F^{(l-1)} \times \cdots \times \nabla^{\times d_{d+1}} F^{(l-1)} \right) \\ = \\ \sum_{d=2}^{D} \sum_{d_1 = 1, d_2 \ldots d_d \in \mathbb{N}}^{d_1 + d_2 + \ldots + d_d = D} \phi^{[d]} \left( F^{(l-1)} \right) \left( \nabla^{\times d_1} F^{(l-1)} \times \nabla^{\times d_2} F^{(l-1)} \times \cdots \times \nabla^{\times d_d} F^{(l-1)} \right) \, . \tag{179}$$

The second contribution can be rewritten as:

$$\sum_{d=1}^{D-1} \sum_{d_1 \ldots d_d \in \mathbb{N}}^{d_1 + \ldots + d_d = D-1} \phi^{[d]} \left( F^{(l-1)} \right) \left( \nabla \times \nabla^{\times d_1} F^{(l-1)} \times \cdots \times \nabla^{\times d_d} F^{(l-1)} \right) \\ = \\ \sum_{d=1}^{D-1} \sum_{d_1 \ldots d_d \in \mathbb{N}}^{(d_1+1) + \ldots + d_d = D} \phi^{[d]} \left( F^{(l-1)} \right) \left( \nabla^{\times (d_1+1)} F^{(l-1)} \times \cdots \times \nabla^{\times d_d} F^{(l-1)} \right) \tag{180} \\ = \\ \sum_{d=1}^{D-1} \sum_{1 < d_1 \in \mathbb{N}, d_2 \ldots d_d \in \mathbb{N}}^{d_1 + \ldots + d_d = D} \phi^{[d]} \left( F^{(l-1)} \right) \left( \nabla^{\times d_1} F^{(l-1)} \times \cdots \times \nabla^{\times d_d} F^{(l-1)} \right) \, .$$

Combining the two sums yields exactly the desired form, which completes the proof of the first case. Lemma G.4 follows directly. □

***Proof - Lemma G.5.***

Proof of the first part:

The number of ways to partition into $d$ distinct sets with $d_1, \ldots, d_d$ objects is:

$$\frac{(d_1 + \cdots + d_d)!}{d_1! \cdots d_d!} = \frac{D!}{d_1! \cdots d_d!} \ , \tag{181}$$

but our sets are not distinct, so we divide by the appropriate coefficient. If the sets are not identical, then they repeat under different permutations, yielding the factor $\frac{1}{d!}$. Summing over all possibilities gives the $D$-th Bell number.

The second part is analogous. □

### G.4 WIDE FCNNS ARE WEAKLY CORRELATED PGDML SYSTEMS

Here we provide a detailed heuristic argument for why wide neural networks are weakly correlated PGDML systems, as described in Lemma G.2.

**Remark G.6.** For this section we assume that the width of the last layer, i.e. the $L$-th layer, is exactly $n_L = 1$. This does not affect any of our results on the system's asymptotic behavior, since $L$ is fixed in $n$, as discussed in Remark C.6.

**Remark G.7.** Throughout this section we use Einstein's summation convention liberally.

We initiate our exploration of wide neural network correlations (and derivatives) by focusing on the most critical one-the kernel-$\mathfrak{C}^1$.

For the final layer $l = L$, the kernel norm is simply:

$$\left\| \mathfrak{C}_{(L)} \right\| = \left| \mathfrak{C}_{(L)} \right| \ . \tag{182}$$

Given that $n_L = 1$, the kernel is a scalar.

Leveraging Lemma G.4, we can construct the $L$-th layer kernel from the components of the preceding layer:

$$\mathfrak{C}^1_{(L)} = \theta_i^{(L,L-1)} \theta_j^{(L,L-1)} \left( \mathfrak{C}^1_{(L-1)} \right)_{ij} + \eta \phi \left( F_j^{(L-1)} \right)^2 \ . \tag{183}$$

Applying Lemma G.1 and the Lipschitz property of $\phi$, we find that the right-hand term satisfies $\eta \phi \left( F_j^{(L-1)} \right)^2 \sim O(1)$. For the left-hand term, Lemma G.4 again yields:

$$\left( \mathfrak{C}^1_{(L-1)} \right)_{ij} = \theta_{ip}^{(L-1,L-2)} \theta_{jq}^{(L-1,L-2)} \left( \mathfrak{C}^1_{(L-2)} \right)_{pq} + \delta_{ij} \eta \phi \left( F_k^{(L-2)} \right)^2 \ . \tag{184}$$

Thus, we obtain an $O(1)$ term and another term depending on the previous layer. Continuing inductively and using the fact that all contributions are symmetric (and hence nonnegative), we conclude that the kernel has asymptotic behavior $O(1)$. **In combination with (G.2), this shows that our system satisfies the criteria of a PGDML system (D.2)!**

Let us now consider a general final correlation $\mathfrak{C}^{D,d}_{(L)}$ with $D \in \mathbb{N}_0$ and $d \in \mathbb{N}$. By Lemma C.3, there exists a vector $v \in S_N$ achieving the norm:

$$\left\| \mathfrak{C}^{D,d}_{(L)} \right\| = \left| \mathfrak{C}^{D,d}_{(L)} \cdot v^{\times D} \right| \ . \tag{185}$$

Applying Lemma G.4, we can express this quantity in terms of correlations from earlier layers. Considering only the first term among the three contributions (the others are treated analogously), and focusing on first correlations, we obtain (up to a factor $\frac{1}{d!}$, while omitting $\frac{1}{D!}$ since we do not distinguish different index arrangements):

$$\left( \phi^{[d+D]} \left( F^{(L-1)} \right) \left( \theta^{(L,L-1)} \right) \right) \times \left( \tilde{\theta}^{(L,L-1)} \right)^{\times d} \cdot \left( \left( \mathfrak{C}_{(L-1)} \right)^{\times d} \times \left( \eta^{\frac{1}{2}} \nabla F^{(L-1)} \right)^{\times D} \cdot v^{\times D} \right) \ . \tag{186}$$

Using (23) and the fact that the layers $L-1$ and $L$ are independent at initialization, we can discard the $\phi$-terms without changing the asymptotic behavior (we discuss the $d!$ factor below):

$$\left(\theta^{(L,L-1)}\right)^{\times(d+1)} \cdot \left(\left(\mathfrak{C}_{(L-1)}\right)^{\times d} \times \left(\eta^{\frac{1}{2}}\nabla F^{(L-1)}\right)^{\times D} \cdot v^{\times D}\right) . \tag{187}$$

When constructing kernels from the preceding layer, each kernel contains two terms (183), resulting in $2^d$ total terms. This factor does not affect the asymptotic behavior, so we consider only the maximal contributions. We focus on the terms built solely from the diagonal contributions and treat the remaining terms by induction:

$$\theta_{i_0}^{(L,L-1)}\theta_{i_1}^{(L,L-1)}\cdots\theta_{i_d}^{(L,L-1)}\left(\delta_{i_0 i_1}\eta\phi\left(F_k^{(L-2)}\right)^2\right)\cdots\left(\delta_{i_0 i_d}\eta\phi\left(F_k^{(L-2)}\right)^2\right)\cdot$$
$$\left(\left(\eta^{\frac{1}{2}}\nabla F_{i_0}^{(L-2)}\right)^{\times D}\cdot v^{\times D}\right) . \tag{188}$$

Since $\eta\phi\left(F_k^{(L-2)}\right)^2 \sim O(1)$, after contracting the Kronecker deltas we obtain an asymptotic behavior of at most:

$$\left(\theta_i^{(L,L-1)}\right)^{d+1}\left(\left(\eta^{\frac{1}{2}}\nabla F_i^{(L-2)}\right)^{\times D}\cdot v^{\times D}\right) . \tag{189}$$

Now, since $O\left(\eta^{\frac{1}{2}}\nabla F_i^{(L-2)}\right) \leq O(1)$, for $D \in \mathbb{N}$ the worst-case asymptotic behavior is:

$$\left(\theta_i^{(L,L-1)}\right)^{d+2} . \tag{190}$$

By proper initialization, this is uniformly bounded for all $d$ by:

$$d!O\left(\frac{1}{\sqrt{n}}\right)^d . \tag{191}$$

Reintroducing the factor $\frac{1}{d!}$ yields:

$$O\left(\frac{1}{\sqrt{n}}\right)^d . \tag{192}$$

If $D = 0$, the factor $\left(\eta^{\frac{1}{2}}\nabla F_i^{(L-2)}\right)^{\times D} \cdot v^{\times D}$ disappears, and we are left with:

$$\left(\theta_i^{(L,L-1)}\right)^{d+1} . \tag{193}$$

For odd $d$, symmetry of $\theta$ still yields $O\left(\frac{1}{\sqrt{n}}\right)^d$. For even $d$, we obtain:

$$O\left(\frac{1}{\sqrt{n}}\right)^{d-1} . \tag{194}$$

**This explains why our system is $\sqrt{n}$-weakly and power correlated.** Nonetheless, one can verify that this term remains negligible in the time deviation as $n \to \infty$.

Of course, there are many other terms besides the first-derivative ones, but they can be treated similarly.

Assume that for layer $l-1$,

$$\phi^{[d']}\left(F^{(l-1)}\right)\mathfrak{C}_{(l-1)}^{D_1,d_1} \times \cdots \times \mathfrak{C}_{(l-1)}^{D_{d'},d_{d'}} . \tag{195}$$

contributes at most:

$$O\left(\frac{1}{\sqrt{n}}\right)^{d \text{ or } d-1} . \tag{196}$$

Using Lemma G.5, and replacing $\phi^{[d']}\left(F^{(l-1)}\right)$ by $d'!$ (as warranted by (23)), we find that the total contribution is bounded by:

$$
\begin{aligned}
\sum_{d'=1}^{D+d} \sum \sum \frac{1}{d'!} \frac{d!}{d_1!\cdots d_{d'}!} \frac{D!}{D_1!\cdots D_{d'}!} d'! \frac{d_1!\cdots d_{d'}!}{d!} \frac{D_1!\cdots D_{d'}!}{D!} O\left(\frac{1}{\sqrt{n}}\right)^{d \text{ or } d-1} \\
\sim 2^{D+d} O\left(\frac{1}{\sqrt{n}}\right)^{d \text{ or } d-1} \sim O\left(\frac{1}{\sqrt{n}}\right)^{d \text{ or } d-1} .
\end{aligned}
\tag{197}
$$

A similar argument shows that products of correlations exhibit the same behavior at the $l$-th layer. Consequently, we can proceed by induction, as in the kernel case, to show that all correlations scale in the same way, thereby concluding our **heuristic proof**.

## G.5 GENERALIZATION BEYOND FCNNS

### G.5.1 TENSOR PROGRAMS

While FCNNs are the prototypical network architecture, numerous other architectures are used in practice, as we discussed in Section 4.2. The tensor programs formalism, as detailed in Yang & Littwin (2021), offers a unified language to encapsulate most relevant neural network architectures by viewing them as composites of global linear operations and pointwise nonlinear functions. This formalism encompasses a wide array of architectures, including recurrent neural networks and attention-based networks. In their work, they demonstrated that any wide network described within this formalism exhibits linearization.

Our weak-correlation approach naturally aligns with the tensor programs framework, simplifying the proof that such networks not only exhibit linearization, but also are weakly correlated PGDML systems. This comes with additional implications, such as deviations over learning and the influence of network augmentation on the linearization rate.

Our proof for FCNNs can be generalized to any wide network described by this formalism, because, similarly to FCNNs, all such systems exhibit a wide semilinear form by definition.

### G.5.2 BEYOND TENSOR PROGRAMS

Given the broad generality of the tensor programs formalism, it is challenging to devise linearizing networks that fall outside its scope. However, we suggest two network-based architectures that demonstrate linearization and, to our belief, fall outside this formalism.

The first is the FCNN in (139), but where each neuron possesses a distinct activation function:

$$
F_i^{(l)} = \sum_{j=1}^{n_{l-1}} \theta_{ij}^{(l,l-1)} \phi_j\left(F_j^{(l-1)}\right) + \theta_i^{(l)} .
\tag{198}
$$

The proof of linearization for this system, assuming $\phi_i$ satisfies the condition 23, parallels our proof for FCNNs.

Not all such systems lie outside the random tensor formalism's purview: if we can represent $\phi_i$ as a function of two distinct inputs-$F_i$ and an external input indexed by $j \in \mathbb{N}$-then we can write:

$$
\forall j = 1, \ldots, n_{l-1} : \phi_j\left(F_j^{(l-1)}\right) = \phi\left(F_j^{(l-1)}, j\right) .
\tag{199}
$$

However, since $\phi$ and its derivatives must remain bounded by some polynomial to fit within the theorems of Yang & Littwin (2021); Yang (2020) for wide neural networks, if the collection $\{\phi_i\}$ is sufficiently diverse, identifying such a $\phi$ may be very challenging or even impossible.

A more definite (albeit synthetic) example of a linearizing network-based system outside the tensor programs realm is:

$$
z(x) = \sum_{i=1}^{n} \theta_i f_i(x) + \sum_{i,j=1}^{n} \theta_i \theta_j g_i(x) g_j(x) \quad g = Af ,
\tag{200}
$$

initialized by $\theta = 0$, where $A$ is a $90°$ rotation matrix acting on the relevant axis as $n \to \infty$, and $f_i$ are chosen as eigenfunctions of an external kernel.

This system can be viewed as an NTK approximation, but with a nontrivial second derivative that is orthogonal to the first. Hence, the system still behaves linearly as $n \to \infty$. It is also not evident how this system can be derived from the tensor programs framework.

While one might contend that this example is artificially contrived, it underscores the existence of weakly correlated, network-based systems that are not captured by the tensor programs formalism.

Furthermore, in line with our discussion in Section 3.3.3, if we manage to identify effective correlations that are beneficial in practice, such systems might find useful applications.

## H    THE CHICKEN AND THE EGG - EXTENDED

In this section, we elaborate on the points made in Section 3.3.3. We begin by explaining why we view derivative correlations as a form of bias in the system.

The simplest way to see the equivalence between weak derivative correlations and an inherent bias in the system is to consider wide neural networks. In our demonstration that wide neural networks exhibit weak derivative correlations (Appendix G), we assumed that the initial distribution of $\theta$ is uncorrelated in the infinite-width limit. If we introduce correlations into $\theta$, then these correlations contribute to the derivative correlations, so that they no longer vanish. The converse also holds: persistent derivative correlations are equivalent to correlations in the initial distribution of $\theta$ in the large-width limit. Such correlations in the initial distribution of $\theta$ indicate an inherent bias in the initial hypothesis function, since they imply a predisposition toward specific regions of parameter space. Therefore, weak derivative correlations can be understood as a manifestation of a weak inherent bias in the initial hypothesis function.

Finite neural networks, by virtue of having a finite number of parameters, are restricted to a small subset of parameter space. Indeed, they can be viewed as infinite networks in which many parameters are set to zero and are not allowed to change during learning. This explains why, even when drawing the initial parameters from an i.i.d. distribution, finite neural networks still exhibit non-vanishing derivative correlations, which are reduced as the width increases.

The equivalence between weak derivative correlations and inherent bias is also reflected in gradient descent (Equation 7). Inspecting this update, one observes two objects that the optimization process attempts to reduce: the derivative of the cost function, $\mathcal{C}'(F(\theta), \hat{y})$, and the gradient of the hypothesis function, $\nabla F(\theta)$. Minimizing the norm of the first term corresponds to learning the data, since this term is reduced when the hypothesis function fits the target function. In contrast, minimizing $\|\nabla F(\theta)\|$ corresponds to the system learning its own structure independently of the data, i.e., expressing bias. To enable gradient descent to substantially reduce the second quantity, the higher derivative correlations must share the same asymptotic behavior as the gradient, as seen in Equation 11 for $1 \leq D$. Thus, weak derivative correlations impede the system's ability to learn its own structure rather than the data, thereby reducing bias.

Furthermore, we argue that this interpretation helps explain both why linear learning is so common and why linear systems are typically outperformed by their nonlinear counterparts. We interpret derivative correlations as an inherent bias, and view linear learning as a consequence of our attempt to minimize this bias. However, in some contexts, certain biases can facilitate learning, as exemplified by explicit and implicit regularization. Thus, weak but nonzero derivative correlations can be beneficial, which helps explain why near-linear learning is often preferable to strictly linear learning. In other words, strictly linear learning takes the weak-correlation principle to an unproductive extreme.

## I    LIMITATIONS, FURTHER DISSECTION, AND GENERALIZATION

In this section, we enumerate the key assumptions that underpin our analysis and propose potential extensions beyond these preconditions. Additionally, we identify potential avenues for further related research.

## I.1 SECTION 2

Our analysis here did not rely on any hidden or nontrivial assumptions, except for those explicitly stated in the tensor definitions. Our findings are generalizable and applicable to any random tensor or variable that depends on some limiting parameter $n \in \mathbb{N}$. Extending our results to any set with a total order is straightforward.

We anticipate that this analytical tool will be beneficial not only for investigating wide neural networks, but also for studying random tensors and variables more broadly, particularly when focusing on their limiting behavior for the reasons delineated in this paper. It upholds several useful algebraic properties C.3, provides a well-defined optimal asymptotic bound for any tensor 2.1, and harmonizes naturally with the notion of "convergence in distribution." Further, owing to its generality, it offers broad applicability. We recommend further exploration of this tool in other settings.

## I.2 SECTIONS 3, 4

### I.2.1 ASSUMPTIONS

1. We assume that $F$, $\mathcal{C}$, and $\phi$ are analytical in their parameters, i.e., they are smooth and their Taylor series converge.

2. All derivatives of $\phi$ are bounded as in Equation 23.

3. Our analysis is restricted to single-batch stochastic gradient descent, and we assume that the training and testing distributions coincide.

4. We assume that $\mathcal{C}$ is convex, i.e., $\mathcal{C}''$ is positive definite.

5. Theorems 3.1 and 3.2, and Corollary 4.1, apply only to PGDML systems, as defined in D.2.

6. Theorem 3.1 and Corollary 4.1 are valid only for sufficiently small $\eta$, of the same order as the $\eta$ required for effective linearization analyses.

7. Corollary 4.1 assumes that the first derivative of $\mathcal{C}$ decays exponentially, and that the second derivative remains bounded over time for the linear solution.

8. The equivalence shown in Theorems 3.1 and 3.2 requires that all derivatives remain fixed. One can, however, state a more nuanced equivalence in which the derivatives change substantially, but the network still behaves linearly, provided this change is orthogonal to $\nabla F(\theta_0)$. Nevertheless, since neural networks satisfy our simpler conditions, we adhere to the version stated above.

### I.2.2 GENERALIZATIONS OF THE ASSUMPTIONS

For Condition 1, while we typically deal with smooth analytical functions, non-continuous hypothesis functions are common, for example with the ReLU activation in neural networks. If the system can be represented as a linear approximation plus a function that is analytical on pieces, and if the non-smooth points form a set of measure zero, then the techniques presented herein can be applied.

Regarding the derivative bound in Condition 2, this requirement is relatively non-restrictive, especially given that $\phi$ should be analytic and that the bound need only hold on an arbitrarily high-probability set, not necessarily on the entire probability space.

Extending the single-input batch gradient descent case in Condition 3 to other batch schemes, such as multiple-input batches or deterministic single-batch GD, is straightforward. This extension amounts to repeating our analysis while adjusting the specifics of the optimization algorithm of interest. The generalization to more complex gradient-based algorithms follows similar lines, albeit with additional nuances.

