# OpenReview forum: "Weak Correlations as the Underlying Principle for Linearization of Gradient-Based Learning Systems"
_ICLR.cc/2026/Conference — ICLR 2026 Poster_

### Official Review · Reviewer_fCjc · 2025-10-29

**Soundness:** 4
**Presentation:** 3
**Contribution:** 4
**Rating:** 8
**Confidence:** 4

**Summary:**

This paper proposes a criterion, that of having weak correlations between the first derivative of the model. and the higher derivatives, and shows that it is equivalent to the model being in a linear/NTK regime where one can take a Taylor approximation around initialization. Thus any model with weak correlation can be proven to convergence exponentially fast, and wide DNNs can be thought as just a special case of this result.

**Strengths:**

This is to my knowledge the first equivalent condition to NTK-type linearization, and offers a pretty novel point of vue. Other conditions such as the ratio of the Hessian norm to the gradient norm were more sufficient conditions. This aligns well with an intuition I (and other) had: proving NTK dynamics by looking at a ball around initialization often yields loose rates, instead one has to leverage the fact that the parameters typically move along directions where the NTK moves little.

**Weaknesses:**

Some parts of the paper are very technical and a bit hard to follow, as well as some parts of the discussion.

Also the criterion is probably very hard to compute in practice because it involves high-dimensional objects (higher order derivatives). It seems that it would be easier in practice to simply compute how much the NTK moves rather than computing these correlation values.

**Questions:**

- It seems that you have missed a paper that is very closely related (it is very similar to Dyer & Gur-Ari 2019): https://arxiv.org/abs/1909.08156 . I wonder how close your condition is to assuming that the higher order terms vanish in the Neural Tangent Hierarchy defined in this paper.

---

> ### Author Response · Authors · 2025-11-28
>
> Thank you for your thoughtful review. We are pleased that you found our paper interesting and important. We are particularly happy to hear that our results aligned with the intuition and ideas of another researcher in the field.
>
> **Regarding your questions and concerns:**
>
> > Some parts of the paper are very technical and a bit hard to follow, as well as some parts of the discussion.
>
> Our subject involves inherently complex concepts, and our intention was to simplify the exposition to make it as accessible as possible. To make our proofs easier to follow, we provided explanations for each step in the appendix, as well as heuristic explanations in the main text after each result is presented. We also aimed to explain the reasoning and intuition behind our analysis. To further improve the readability of the Appendix, we will add short introductory overviews at the beginning of the relevant sections in the appendix, similar to the general explanations we already include in the main text for the theorems and corollary.
>
> > Also the criterion is probably very hard to compute in practice because it involves high-dimensional objects ... It seems that it would be easier in practice to simply compute how much the NTK moves rather than computing these correlation values.
>
> That said, we have developed a more practical way to compute the correlations numerically. One does not need to evaluate the full $D$-th derivative of the network to compute the $D$-th correlation. Instead, it is possible to use the following trick:
> $$
> C\_{\alpha\_{1}\dots\alpha\_{D}}\left(x\_{0},x\_{1}\dots x\_{D}\right)\_{i\_{0},i\_{1}\dots i\_{D}}=\partial\_{\alpha\_{1}}\cdots\partial\_{\alpha\_{D}}F\_{i\_{0}}\left(\theta\_{0}+\alpha\_{1}\nabla F\_{i\_{1}}\left(\theta\_{0},x\_{1}\right)+\dots+\alpha\_{D}\nabla F\_{i\_{D}}\left(\theta\_{0},x\_{D}\right),x\_{0}\right)\bigg|\_{\alpha\_{1}=\dots=\alpha\_{D}=0}
> $$
>
> which requires only to compute the gradients. This makes the procedure linear in the order of the correlation, and avoids the need to compute high-order derivatives directly.
>
> Thank you for emphasizing the importance of this nuance. We will include this explanation in the next revision of the text.
>
> > It seems that you have missed a paper that is very closely related... how close your condition is to assuming that the higher order terms vanish in the Neural Tangent Hierarchy defined in this paper.
>
> Thank you for pointing out this important paper. It is indeed closely related to our work.
>
> In the specific setting they consider, i.e., gradient flow over the specific kind of wide neural networks they allow, their kernels can be expressed as linear combinations of our correlations. However, our result is more general, as it does not rely on the specific structural assumptions of any wide neural network. The most immediate benefit of that is that our framework can be applied to more learning systems that their approach does not capture.
>
> More fundamentally, by not restricting ourselves to neural networks, and instead introducing these new correlations, we are able to obtain not only a sufficient condition for linearization, but, as you mentioned, an equivalence. This perspective is also key to our interpretation presented in Section 3.3.3. In addition, our results apply to gradient descent rather than gradient flow alone, making them more practical for real-world research. we will make sure to discuss that in the revised related-work section.
>
> Thank you for your thoughtful review. We are happy that a researcher who has independently thinking on similar questions found our paper interesting and impactful, and that the perspectives we offer resonated with you.

---

### Official Review · Reviewer_ERGz · 2025-10-30

**Soundness:** 3
**Presentation:** 1
**Contribution:** 2
**Rating:** 4
**Confidence:** 3

**Summary:**

This paper tries to connect the linearity of training wide neural networks with the weak correlations between the first and higher-order derivatives. The authors propose a novel formalism for this purpose by investigating the asymptotic behavior of random tensors, which might be generalized to other cases of machine learning.

**Strengths:**

The framework of this paper is generally novel. The goal of this paper is clear and straightforward. To this end, the authors are able to develop a sophisticated asymptotic theory for random tensors, which might be of independent interest. This formalism is roughly applied consistently across the proofs, providing a uniform treatment of various architectures.

**Weaknesses:**

1. I think the presentation is obviously under the bar of ICLR, which sometimes makes the paper challenging to read. Below I list some examples:

   - line 278 "under the conditions described above". I'm confused about the exact conditions mentioned here: it seems that there is not any condition above this line. Similar issues exist in Theorem 3.2 line 298.

    - line 279 "sufficiently small learning rate $\eta < \eta_{the}$". How is this $\eta_{the}$ obtained? And how small it should be?

    - line 294, I think a "for example" should not be stated in a formal Theorem.

    - line 405-407 "then exists some 0< T, such as for every s =1 ... S, if:". I'm confused about the statement:

        - $0 < T$ is obvious, why exists?

        - "such as for every $s$": Why "such as" here? It seems that "such as" should not be in a Corollary.

         - Again, how is $\eta_{cor}$ obtained here? Is it a same one as $\eta_{the}$ in Theorem 3.2?

2. The authors do not sufficiently justify that existing tools are inadequate for deriving the main results, which makes the motivation of designing a new formalism a bit unclear. In addition, the overwhelming technical machinery in fact raises a barrier for readers to appreciate the methodology, and I'm only able to roughly read the proofs.

3. The authors make a new and even bold claim, yet the empirical validation is far from sufficient to support this new claim.

4. Indeed, the framework is general. However, the core insight seems to be a deep reformulation of existing knowledge (lazy training, NTK, infinite-width limits). From this perspective, I think the contribution is limited.

**Questions:**

Please see the first point of the Weaknesses.

---

> ### Author Response · Authors · 2025-11-28
>
> Thank you for your review, questions, and comments.
>
> **About the presentation:**
>
> We appreciate your comment regarding the presentation. Our subject involves inherently complex concepts, and our intention was to simplify the exposition to make it as accessible as possible. Based on your feedback, we will revise the paper to clarify the technical details and improve readability, particularly regarding the specific points you have raised.
>
> > ..."under the conditions described above". I'm confused about the exact conditions mentioned here...
>
> We were referring to the conditions detailed in Section 3. We will revise this phrasing to make it clearer, and add a reference to the appendix where these conditions would be listed in details.
>
> > ...“sufficiently small learning rate.” How is this $\eta_{\mathrm{the}}$ obtained?...
>
> and
>
> > I think a “for example” should not be stated in a formal Theorem.
>
> As stated at the end of the theorem, this learning rate is the smallest learning rate required to ensure that all correlations remain bounded, i.e., that none of them diverge after a single gradient descent step. This is a necessary condition for any practical gradient descent based algorithm implementation. Any system that does not satisfy this condition will diverge within only a few training steps, which makes it a very mild assumption. As we agree that this point is important to deserve a clearer explanation in the main text, we will remove the “for example” from the theorem, and add a dedicated paragraph clarifying how $\eta_{\mathrm{the}}$ is defined and why this condition is needed.
>
> > $0 < T$ is obvious, why exists?
>
> and
>
> > Why “such as” here? It seems that “such as” should not be in a Corollary.
>
> The condition appears in the corollary because we assume that for $s = 1, \ldots, S$, the linear solution converges exponentially according to this typical learning time $0<T$. In the appendix, we show that for any fixed $S\in\mathbb{N}$, such a $0<T$ must exist as long as the system doesn't diverge, and that in typical settings, this $T$ will be relatively large. Because of space constraints, we stated this more briefly in the main text, but we agree that it is important enough to clarify explicitly. In the revised version, we will add a paragraph explaining the existence and role of $T$, and adjust the phrasing of the corollary.
>
> > ...how is $\eta_{cor}$ obtained here?...
>
> As explained at the end of the corollary, and in more detail in the appendix, $\eta_{cor}$ is the standard NTK threshold on the learning rate required for stable training. It is typically of the same general order of magnitude of $\eta_{the}$.

---

> ### Author Response · Authors · 2025-11-28
>
> **About your other concerns:**
>
> > The authors do not sufficiently justify that existing tools are inadequate for deriving the main results....
>
> All of our results are either impossible to derive using existing tools, since they are formulated within our new formalism, or they rely directly on properties of our formalism to simplify their proofs. Theorems 3.1 and 3.2, for example, are expressed in our framework, and therefore could not have been obtained with previously available techniques. These theorems constitute the first structural criteria for the linearization of any gradient descent-based learning system, independent of the underlying neural network architecture. This level of generality allows us to state the results as true equivalences rather than merely sufficient conditions, a substantial advantage over existing approaches.
>
> These special properties also enabled us to establish several results that had not been proven before, including the bound on network linearization as a function of the activation (Section 4.2), and the characterization of its deviation over time (Corollary 4.1). While it is of course impossible to prove that these results could not, in principle, be obtained by other means, we believe that our approach renders them accessible precisely because it analyzes linearization directly threw the correlations, rather than indirectly through other frameworks such as Gaussian process approximations. The fact that our formalism yields these results cleanly, and that we were the first to obtain them, supports the naturalness and usefulness of the perspective we introduce. Moreover, our interpretation provides an additional conceptual viewpoint that may facilitate further developments, as discussed in Section 3.3.3. For these reasons, we believe our formalism is likely to open the door to further new results.
>
> Thank you for emphasizing the need to clarify this important point. We will expand this discussion in the next version of the paper and better highlight the comparative contribution of our approach.
>
> > ...the overwhelming technical machinery in fact raises a barrier for readers to appreciate the methodology, and I'm only able to roughly read the proofs...
>
> Our subject involves inherently complex concepts, and our intention was to simplify the exposition to make it as accessible as possible. To help readers follow the proofs, we included explanations for each step, along with heuristic discussions in the main text after each result. To further improve readability, we will add brief introductory overviews at the beginning of the relevant appendix sections, in the same spirit as the heuristic explanations already provided in the main text.
>
> > The authors make a new and even bold claim, yet the empirical validation is far from sufficient to support this new claim.
>
> As our work is theoretical, and all of our results are established through rigorous proofs, we believe that the standards for empirical demonstration can be lower than for works that are primarily empirical. Nonetheless, to complement the theory and make its implications more concrete, we are preparing a more extensive empirical study. These additional experiments will be incorporated into the next version of the paper.
>
> > ...the core insight seems to be a deep reformulation of existing knowledge...
>
> Our work provides a new and more general way to understand when and why learning systems linearize. We establish a full equivalence between linearization and a local property of the system, an equivalence that applies not only to neural networks, but to any gradient descent-based learning system. This is, to our knowledge, the first structural criterion of this form.
>
> This perspective also leads to concrete practical results. For example, in Section 4 we use this framework to derive new bounds on network linearization as a function of the activation and to characterize how deviations from linearization evolve over time. These results were not available before and follow naturally from the formalism we introduce.
>
> In addition, the connection we draw between linearization and implicit bias in Section 3.3.3 offers a new conceptual angle that we believe can support further theoretical progress.
>
> Taken together, our contributions include both new practical insights and new conceptual tools. We view this type of theoretical work, similar in spirit to other highly cited foundational papers in the field, as valuable for advancing the broader understanding of machine learning and for opening the path to future discoveries.
>
> We thank you again for your review, and we hope that our responses address your concerns regarding the overall quality and contribution of our work. Together with the revisions we plan to make based on your feedback and that of the other reviewers, we hope you will consider increasing your score.

---

### Official Review · Reviewer_C8ci · 2025-10-31

**Soundness:** 2
**Presentation:** 3
**Contribution:** 2
**Rating:** 4
**Confidence:** 3

**Summary:**

This paper offers a nice unified explanation for NTK‑style linearization (weak derivative correlations) and a careful asymptotic calculus to make that explanation precise. It provides some helpful intuition for understanding the driver of the lazy regime vs. feature‑learning regimes.

**Strengths:**

- Clear unifying idea. The paper puts forward weak derivative correlations, small correlations at initialisation between the first and higher‑order parameter derivatives, as the underlying mechanism for NTK‑style linearization, with precise definitions $(C_{D,d}$ and two equivalence theorems (Theorems 3.1 and 3.2) tying correlation decay to linearised dynamics and learning‑rate scaling. This gives a compact, testable lens on “lazy training.”
- Technical framework. Section 2 builds a random‑tensor asymptotics calculus (subordinate tensor norm + stochastic big‑O, uniform bounds), and proves existence/uniqueness of a tight upper bound.
- Deviation‑over‑time statement. Corollary 4.1 bounds the SGD deviation $F- F_{\text{lin}}$ by $O(1/m(n))$ over (finite) time under an exponential NTK‑phase contraction assumption, making the linearisation guarantee feel more operational.
- Attempt at architectural breadth. By leaning on tensor‑programs, the authors argue many wide architectures satisfy weak correlations (with rates tied to activation derivatives, Equation 22), offering a route to reason about how architectural choices and learning‑rate reparametrization $\eta \mapsto r(n)\eta$ push systems toward or away from linearisation.

**Weaknesses:**

- All experiments use tiny subsets of MNIST, CIFAR‑10, and Fashion‑MNIST; fully connected nets; MSE loss; and NTK‑normalised learning rate with long training (1,000 epochs). These datasets/architectures in this setup typically do not require rich feature learning and are well known to be close to the lazy/NTK regime already. Thus, showing that the relative discrepancy $|f - f_{\text{lin}}|$ shrinks with width (Figure 1) and that estimated 2nd/3rd‑order correlation proxies decrease (Figure 2-3) does not validate the central predictive claim in regimes where feature learning matters. That is - nowhere do we empirically validate that networks that are known to feature learn do not also have weak correlations between the first and higher order derivatives. I appreciate that it is hard to compute the derivatives with large amounts of data (where feature learning typically happens) but there are toy models that show feature learning with relatively small datasets (e.g. sparse parity, multi-index, staircase functions).

**Questions:**

1. Main question (feature learning regime). Can you evaluate on feature‑learning regimes (even on small datasets as described in weaknesses), and show that your correlation diagnostics measured at initialisation predict the gap between finite‑width training and NTK?
2. Learning‑rate scaling predictions. Theorem 3.2 claims reparametrising $\eta \mapsto r(n)\eta$ modulates linearity. Can you add experiments that sweep (r(n)) with width to confirm the predicted $O(r(n)/m(n))$ deviation and the $O((1/\sqrt{m(n)})^d)$ decay of $C_{D,d}$?

---

> ### Author Response · Authors · 2025-11-28
>
> Thank you for your review. We are pleased that you found our paper clear, interesting, and practical.
>
> We will start by answering your question and addressing your concern about the relevance of our results in the feature-learning regime, as the two are closely related.
>
> > ...nowhere do we empirically validate that networks that are known to feature learn do not also have weak correlations between the first and higher order derivatives...
>
> and
>
> > ...Can you evaluate on feature-learning regimes (even on small datasets as described in weaknesses), and show that your correlation diagnostics measured at initialisation predict the gap between finite-width training and NTK?
>
> Thank you for raising this important discussion.
>
> Our main theoretical and empirical results precisely characterize the transition between the regime where the kernel still changes meaningfully and the “lazy training” regime where it remains stable. While we agree that the architectures and tasks we used are simple, we also evaluated them at small widths, where the networks are still far from the NTK limit. You can see in our experiments that the difference between the nonlinear network and its linearization is substantial at the smaller widths we examined, and the correlations we measure are indeed significantly larger than those observed at larger widths. We will highlight this important point more clearly in the revised version of our manuscript.
>
> > Can you add experiments that sweep $r(n)$ with width to confirm the predicted $O\left(\frac{r(n)}{m(n)}\right)$ deviation...
>
> Thank you for the suggestion. We agree that this is a valuable idea, and we will include this experiment in the camera-ready version, together with the other experiments we plan to present.
>
> Thank you again for your review. We hope that in light of our responses, and the revisions we plan to make based on your feedback and that of the other reviewers, you will consider increasing your score.

---

### Author Response · Authors · 2025-11-28

Dear Area Chair,

Due to the unfortunate circumstances regarding the OpenReview leak, we understand that we can no longer engage in discussion with the reviewers to alleviate their concerns or update scores. We hope that our previously submitted rebuttal effectively addresses the points they raised.

---

### Author Response · Authors · 2025-12-03

Dear Area Chair,

Thank you for taking over the handling of our submission, and for your efforts in guiding it through the recent transition.

We would like to inform you that we have addressed all of the major concerns raised in the reviews. The corresponding revisions in the manuscript are highlighted in blue for ease of reference. We are currently running the additional experiments suggested in the reviews, and we will incorporate the final results by the camera-ready deadline.

We appreciate your time and consideration, and we hope that the revised version satisfactorily addresses the reviewers’ concerns.

---

### Meta-Review · Area_Chair_qPip · 2025-12-26

**Summary:**

This paper develops a formal criterion for NTK-style linearization based on weak correlations between first- and higher-order derivatives, and proves equivalence results linking this condition to dynamics remaining close to the linearized model. The framework is technically careful, gives a unifying perspective on lazy training, and appears broadly compatible with many architectures.

Reviewers appreciated the conceptual clarity and the rigor of the analysis, and viewed the equivalence perspective as a meaningful refinement over earlier sufficient-condition results. At the same time, concerns remain: the presentation is dense and sometimes opaque; the empirical support is confined to settings already close to NTK behavior; and it is not yet clear how practically useful the proposed criterion is, beyond explanatory value.

Overall, this feels like a solid, theory-driven contribution that helps organize and clarify an active line of work, even if its immediate impact and empirical breadth are limited. I recommend a weak accept as a thoughtful theoretical paper that is worth being part of the literature, with encouragement to improve exposition and to probe regimes where genuine feature learning occurs.

**Reviewer Concerns:**

Addressed:
The rebuttal clarified several presentation issues (definitions, assumptions, role of learning-rate thresholds), and argued more clearly why the new formalism is needed relative to existing NTK tools. It also provided better intuition for the equivalence results and promised additional experiments (including learning-rate sweeps).

Still outstanding:
The empirical validation remains narrow and does not convincingly probe feature-learning regimes. It is still unclear how practical the correlation criterion is, beyond explanatory value. Finally, the paper remains technically dense, and the broader impact beyond reformulating existing NTK intuition is not fully demonstrated.

**Reviewer Scores:**

C8ci (4) – Clarifications helped, promised experiments are reassuring -> likely unchanged (maybe +1 at most).
ERGz (4) – Presentation/addressed points, but breadth/impact concerns remain -> likely unchanged.
fCjc (8) – Already strongly positive, rebuttal consistent with prior view -> unchanged.

---

### Decision · Program_Chairs · 2026-01-26

Accept (Poster)